# Different environmental variables predict body and brain size evolution in *Homo*

Manuel Will [1,6 ✉], Mario Krapp [2,3,6], Jay T. Stock [4,5] & Andrea Manica [2]

Increasing body and brain size constitutes a key macro-evolutionary pattern in the hominin lineage, yet the mechanisms behind these changes remain debated. Hypothesized drivers include environmental, demographic, social, dietary, and technological factors. Here we test the influence of environmental factors on the evolution of body and brain size in the genus *Homo* over the last one million years using a large fossil dataset combined with global paleoclimatic reconstructions and formalized hypotheses tested in a quantitative statistical framework. We identify temperature as a major predictor of body size variation within *Homo*, in accordance with Bergmann's rule. In contrast, net primary productivity of environments and long-term variability in precipitation correlate with brain size but explain low amounts of the observed variation. These associations are likely due to an indirect environmental influence on cognitive abilities and extinction probabilities. Most environmental factors that we test do not correspond with body and brain size evolution, pointing towards complex scenarios which underlie the evolution of key biological characteristics in later *Homo*.

[1] Department of Early Prehistory and Quaternary Ecology, University of Tübingen, Tübingen, Germany. [2] Evolutionary Ecology Group, Department of Zoology, University of Cambridge, Cambridge, UK. [3] GNS Science, Lower Hutt, New Zealand. [4] Department of Anthropology, Western University, London, ON, Canada. [5] Department of Archaeology, Max Planck Institute for the Science of Human History, Jena, Germany. [6] These authors contributed equally: Manuel Will, Mario Krapp. ✉email: manuel.will@uni-tuebingen.de

Body and brain size are two essential biological traits of a species' adaptive strategy. Key discussions in hominin evolution center around these characteristics, including subsistence strategies, life history variation, energetics, and the origin, diversification, and geographic expansion of our genus. Recent studies have refined and expanded earlier estimates of body and brain size variation across the hominin lineage[1], elucidating taxonomic, temporal, and geographical patterns[2–7]. Throughout the past 4 million years (Ma), human evolution is broadly characterized by a trend of increasing body mass and stature, with an even greater relative increase in brain size, associated with changes in behavior, diet, cognition, and demographic expansion. The past 2 million years have seen an increase in estimated body size among most *Homo* species from an average of ~50 to ~70 kg, and a particularly rapid rise in absolute and relative brain size between 800 and 200 thousand years (ka) ago. The underlying mechanisms for these changes, however, remain understudied and contentious. Many different factors have been proposed to drive the evolution of hominin body and brain size, including climatic and ecological[8–13], dietary[14,15], competitive and social[16–18], and cultural and technological factors[19–21]. Yet, formal tests of these hypotheses have rarely been conducted.

Concerning environmental factors, recent studies have reached discordant conclusions on the importance of latitude, temperature, net primary productivity (NPP), and their variability, attributing them either a major[8,12,22,23] or minor role[2,13] in driving changes in body and brain size. These studies have relied on semi-quantitative, qualitative, and verbal assessments of environmental conditions, such as plotting brain and body size estimates by latitude or on global oxygen isotope curves ($\delta^{18}O$), sea-surface temperatures, and marine isotope stages[2,8,12,13]. The nearly exclusive reliance (but see ref. [24]) on global averages is particularly problematic, as climate change has different impacts in different geographical areas and across time, even on geological time scales[25]. Consequently, more empirical work on the relation between hominin body/brain size and paleoenvironmental variables[18], while also considering sampling issues for the fossil record[26], has been called for.

Here we test the influence of environmental factors on the evolution of body and brain size in the genus *Homo* over the past ~1 Ma. In a first step, we build a conceptual framework by formalizing four environmental hypotheses that relate climatic variation to body or brain size. We test the relationship between body/brain size and local climatic variables in accordance with these hypotheses. Our body ($n = 204$) and brain size ($n = 166$) estimates come from individual fossils of *Homo* (see "Methods") distributed throughout the Old World and ranging from ca. 1.0 to 0.01 Ma (Fig. 1). We divide this dataset into three taxonomic units: Mid-Pleistocene *Homo*, *Homo neanderthalensis*, and Pleistocene *Homo sapiens* (see "Methods"). The environmental information for each individual data point (i.e., geographical location and age of each fossil specimen) comes from a climate emulator[27] that takes into account long-term, glacial–interglacial climate variation, caused by changes in the Earth's orbit around the Sun (Milankovitch cycles)[28] and in greenhouse gases, such as $CO_2$.

We start by formalizing the environmental hypotheses. The most well-known hypothesis for a phenotypic relationship to the environment is Bergmann's[29] rule, predicting a larger body size in colder environments and smaller body mass in warmer environments. The commonly accepted logic behind Bergmann's rule is that a large body size buffers individuals against the challenges of cold climates, either in terms of thermoregulation and/or resource storage. Contemporary humans broadly fit this pattern (e.g. refs. [30,31]). In the context of primate and hominin body and

brain size evolution, further hypotheses have focused not only on absolute temperature and correlates such as latitude, but also on biomes, precipitation, NPP, and the seasonal, intra-annual or millennial variation in these variables[2,8,11–13,22,23,31–36]. It is difficult to reconcile these hypotheses because they have often been framed in ambiguous terminology, exhibit overlap with one another, provide conflicting predictions, and apply to different timescales.

All these hypotheses are united by the presence of environmental challenges faced by hominin species, which need to be overcome by directional adaptations in body and brain size. Despite differences in detailed mechanisms, the main challenge is the stress resulting from either extreme environmental states (synchronic: dry and arid conditions, low resource availability) or unpredictability (diachronic: habitat instability and fragmentation, resource fluctuation). Furthermore, different hypotheses address different temporal scales. Hypotheses to explain change in body/brain sizes over short-term scales emphasize phenotypic plasticity or natural selection as the result of environmental challenges posed on individuals either throughout their lifespans or across few generations. The long-term scale focuses on environmental challenges posed over (many) millennia, with long-term fluctuations leading to the extinction of whole populations or lineages that lack traits to buffer environmental stress during challenging times. In this latter scenario, some phenotypes might evolve periodically but they eventually go extinct, thus leading to the persistence only of phenotypes that can withstand the more challenging periods. Disentangling these two scales among previously proposed hypotheses is relevant from an analytical perspective, as different climatic variables are required to test their predictions (see Table 1).

Given these general considerations, we formulate four broad hypotheses for the evolution of brain and body size that are relevant within the context of hominin evolution, though others are conceivable. The main interest of these hypotheses lies in explaining the evolution of larger body and brain size since the main trend among hominins is one of increase through time. The formulated hypotheses are distinguished from each other by the type of mechanism that underlies them, rather than a specific environmental variable.

Environmental Stress Hypothesis: Larger brain and body sizes are found in colder, drier, and nutrient-poorer environments as cognitive and physiological buffers against these circumstances. Environmental stress is countered by adaptive mechanisms to cope with greater environmental extremes over short-term scales by increased behavioral or cognitive flexibility (brain), by higher mobility and reduced vulnerability to predation, or phenotypic adaptation through plasticity or natural selection (body).

Environmental Constraints Hypothesis: Larger body and brain sizes are found in environments of higher nutritional sufficiency, allowing for their energetically demanding growth and maintenance without reduced fitness over short-term scales. Conversely, habitats with low-resource availability constrain body and brain size, decreasing mortality risks from potential food shortages.

Environmental Variability Hypothesis: Larger brain and body sizes reduce extinction risk in the presence of environmental variability, habitat instability, and fragmentation on intergenerational, multi-millennia scales. Increased behavioral flexibility (brain) or physiological responses and higher migration rates (body) will buffer against greater magnitudes of variability and unpredictability.

Environmental Consistency Hypothesis: Larger body and brain sizes are found in environments that are consistent on intergenerational, multi-millennial scales as long-term nutritional sufficiency is ensured. Conversely, higher long-term variability in

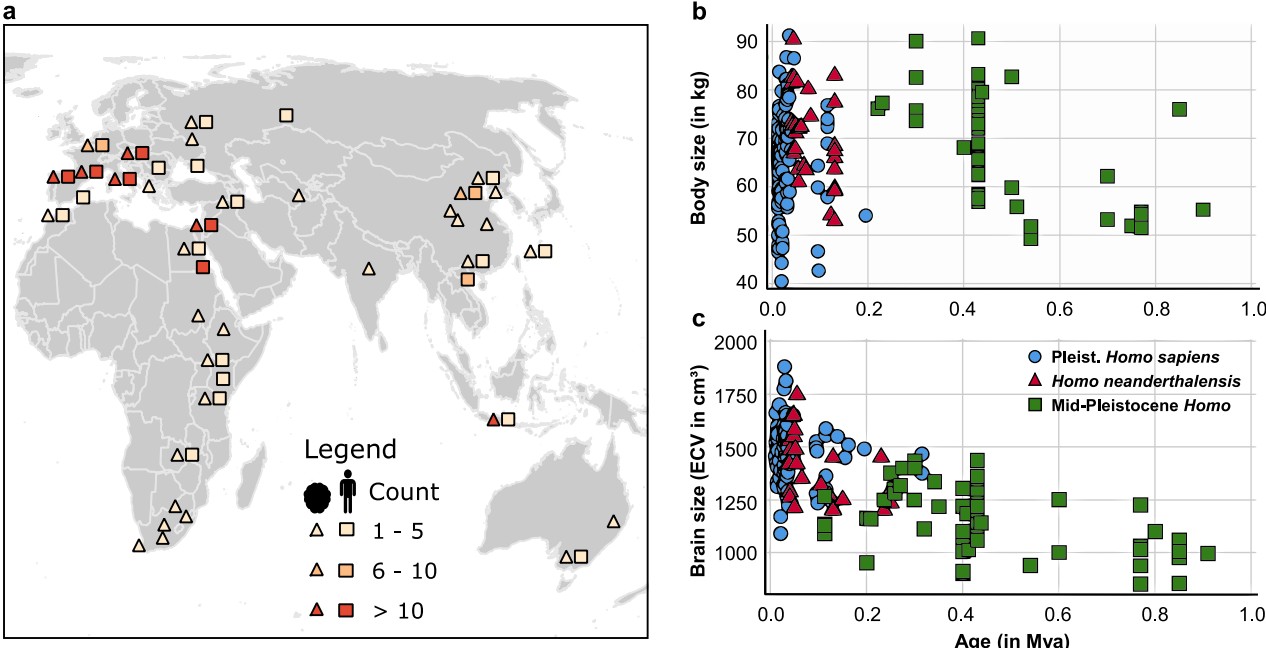

**Fig. 1 Overview of body and brain size datasets for *Homo* used in this study. a** Location and sample size (*n*) of body (squares) and brain size (triangles) estimates for individual *Homo* fossils used in this study (exact locations per specimen can be found in interactive webmap in Supplementary Note 1). **b** Time series for individual body size estimates (*n* = 204) with taxonomic attribution; **c** time series for individual brain size (*n* = 166) estimates with taxonomic attribution. Source data are provided in Supplementary Data 1 and 2.

**Table 1 Formalized environmental hypotheses for changes in brain and body size in *Homo*.**

| Hypothesis | Description | Environmental variable | Expectation brain/ body size |
|---|---|---|---|
| Environmental Stress Hypothesis | Colder, drier, and nutrient-poor environments = larger body/brain size | Mean annual temperature (MAT) Mean temperature of coldest quarter Mean precipitation of driest quarter Net primary productivity (NPP) | Negative correlation |
| Environmental Constraints Hypothesis | Sufficient nutrition required for larger body/brain size | Mean annual precipitation (MAP) Net primary productivity (NPP) Mean precipitation of driest quarter | Positive correlation |
| Environmental Variability Hypothesis | Environmental variability = larger body/ brain size | MAPvar10: $\log[1 + \sigma_{10ka}(MAP)]$ MATvar10: $\log[1 + \sigma_{10ka}(MAT)]$ NPPvar10: $\log[1 + \sigma_{10ka}(NPP)]$ Mean temperature of coldest quarter (var10): $\log[1 + \sigma_{10ka}(\text{Mean Temperature of coldest quarter})]$ Mean precipitation of driest quarter (var10): $\log[1 + \sigma_{10ka}(\text{mean precipitation of driest quarter})]$ | Positive correlation |
| Environmental Consistency Hypothesis | Consistent climate = larger body/brain size; variable climate = smaller body/ brain size | MAPvar10: $\log[1 + \sigma_{10ka}(MAP)]$ MATvar10: $\log[1 + \sigma_{10ka}(MAT)]$ NPPvar10: $\log[1 + \sigma_{10ka}(NPP)]$ Mean temperature of coldest quarter (var10): $\log[1 + \sigma_{10ka}(\text{mean temperature of coldest quarter})]$ Mean precipitation of driest quarter (var10): $\log[1 + \sigma_{10ka}(\text{mean precipitation of driest quarter})]$ | Negative correlation |

Note the different temporal scales of the environmental variables. The term $\log[1 + \sigma_{10ka}()]$ refers to the logarithm of the 10,000 year-running standard deviation of the respective variable (with plus 1 to avoid the logarithm from becoming infinite).

resource availability constrains body and brain size, with an extinction filter against large and energetically expensive phenotypes, which might evolve and survive for short periods but fail to persist in the long term.

We selected environmental variables that are proxies for the mechanisms invoked for each hypothesis, with a clear prediction of the direction of the relationship (Table 1). For each of the body and brain size measurements available (see Fig. 1 and

Supplementary Data 1 and 2), we reconstructed climatic variables for the appropriate locations and time periods using a global climate model emulator (see "Methods")[27]. We then fitted linear models linking body or brain size to each of these climatic variables, formally testing their relationship and their sensitivity to uncertainties in chronometric ages and climate variables (see "Methods"). Given the many biases and uncertainties of the hominin fossil data, for each environmental hypothesis, we first

estimate the power of such a dataset to detect relationships of different environmental effects via the generation of 1000 synthetic datasets for all climate variable associations with body and brain size (see "Methods"). In a second step, we test and discuss the four environmental hypotheses for body and brain size among the *Homo* lineage with the real fossil datasets.

In this study, we show that different environmental variables predict body and brain size in the genus *Homo* over the past 1 Ma. Temperature is a major predictor of body size variation, with larger-bodied individuals consistently occurring in colder climates. These results are in accordance with Bergmann's rule and support the Environmental Stress Hypothesis. Brain size correlates with NPP of environments and long-term variability in precipitation, corresponding with the Environmental Consistency Hypothesis. These variables, however, account for only small amounts of the observed variation in brain size. Other environmental factors that we test are not associated with body and brain size evolution in later *Homo*. Our work suggests that past climatic variation underlies, in part, the evolution of key biological characteristics in Pleistocene *Homo*. A significant proportion of variation remains unexplained by environmental factors, requiring further studies that incorporate tests of social, dietary, and technological drivers by explicit hypotheses with statistical analyses.

## Results

**Approach of power analysis and linear regressions**. Given the sparse nature of the fossil record and dating uncertainties, it is important to assess the power of our analyses given the limitations of the available datasets (e.g., ref. [26]). We designed a power analysis to assess the strength of relationships that we could conceivably detect, as well as which variables (and thus hypotheses) we had sufficient information to test. This information is essential to interpret negative results, as a non-significant relationship is only informative if there is sufficient power to detect the effect size of interest. Our power analysis accounts for uncertainties of dating, climate reconstructions, and the intra-population variability of brain or body size. Using a linear model (see "Methods"), we generated 1000 synthetic fossil brain and body size datasets assuming a weak, medium, or strong relationship with each of the climate variables (see "Methods" for explanation of these terms). Finally, to avoid a few oversampled fossil sites that contain multiple specimens with the same age driving the results, each synthetic dataset was randomly thinned by only retaining one specimen for any given location–time combination. This process was repeated to generate 100 randomly thinned versions of each of the 1000 synthetic datasets.

For each climate variable and thinned dataset, we fitted three linear models: one for taxonomic differences only (LM-T)—which can be regarded as our null model—one for taxonomic differences plus a climate effect (LM-TC), and one for taxonomic differences plus a taxon-specific climate effect (LM-T*C), which allows for a different slope of the relationship for each taxon (see "Methods" for details). We then compared the explanatory power of these models using the Akaike Information Criterion (AIC)[37], estimating the difference in AIC between the alternative models and the null model. A positive difference (ΔAIC > 0) implies that the alternative model is the better model, but we chose a more conservative AIC difference of 2 to yield more robust results. The power of our analysis in recovering a relationship of a given strength between size and a climatic variable was then defined as the proportion of datasets for which a (hypothetical) climatic effect could be detected (i.e., the AIC values of LM-TC and LM-T*C are ≥2 compared to the AIC of the null model, LM-T). A power of >80% is considered as adequate when designing

experiments; thus, a negative result for any variable with a power >0.8 can be considered informative in dismissing a certain hypothesis.

For the fossil data analysis, we then used a similar approach as the one employed for the synthetic data. We generated 1000 thinned datasets by taking random samples accounting for the uncertainty of the chronometric ages and climate reconstructions for each body/brain size estimate. This allowed us to explore the sensitivity of our results to these uncertainties.

**Power analysis of synthetic data**. Given a fossil dataset such as ours, the power analysis suggested that we should be able to detect (power >80%) a *strong* relationship for body size with temperature of the coldest quarter (defined as an effect size of ±0.37%/°C, see "Methods" for details), if such an association exists for the fossil data (Fig. 2). Similarly, we had the power to detect a strong relationship with mean precipitation of the driest quarter (hypothetical effect size ±3%/(mm/a)). These variables underpin the Environmental Stress and the Environmental Constraints Hypothesis. Thus, a negative result for analyses of the real data involving these variables could be considered as evidence against large effects; on the other hand, power was limited to detect medium and small effects, which thus could be missed when analyzing the real data (hypothetical effect sizes can be found in Supplementary Table 3). Irrespective of the size of the effect, we had limited power for variables that are linked to the Environmental Variability and Environmental Consistency Hypothesis, suggesting that any negative results from the analyses of the real data should not be taken as strong evidence against these hypotheses. For brain size (Fig. 2), we had good power (>80%) for all 5 short-term environmental variables for our hypothetical medium and strong effect sizes. As observed for body size, the analyses had low power to detect associations with long-term (10 ka) measures of climatic variability.

**Analysis of fossil data**. For the body size dataset, we detected an association with mean annual temperature (MAT; Fig. 3a and Table 2): larger individuals are found in places where MAT is lower. The LM-TC model had the strongest support based on a lower AIC relative to the null model LM-T (ΔAIC ≥ 2 in 99% of thinned datasets) implying that the response of body size to temperature was the same across the three taxonomic units with a median body size increase of 0.87% per degree of cooling (Supplementary Table 1). For example, a 2 °C cooling in MAT would be associated with a body size increase of 1 kg for individuals with a weight of 60 kg (0.87%/°C × 2 °C × 60 kg = 1 kg). This relationship was 1.5 times our hypothetical *strong* effect size (0.57%; Supplementary Table 3), for which we already had good power (Fig. 2). Effect sizes and $R^2$ values for the fossil and the power analysis in natural units can be found in Supplementary Tables 1 and 2 and in Supplementary Fig. 1. We also identified a relationship (−0.62%/°C) with the mean temperature of the coldest quarter (Fig. 3b, Table 2, and Supplementary Table 1). Again, this effect size is much larger (~1.6 times) than the hypothetical *strong* effect (0.37%) we had assumed in the power analysis. To put these effects into context, the null model that only accounted for differences among the taxonomic units (LM-T) explained ~5% of variance in body size, while MAT or temperature of the coldest quarter add another 15–16% of explained variance. In other words, the effect of MAT accounted for three times the amount of variation explained by differences among taxonomic groups. No other tested variable was a good predictor of body size (Table 2 and Supplementary Figs. 2–5), but we note that our power was limited for these other variables (Fig. 2).

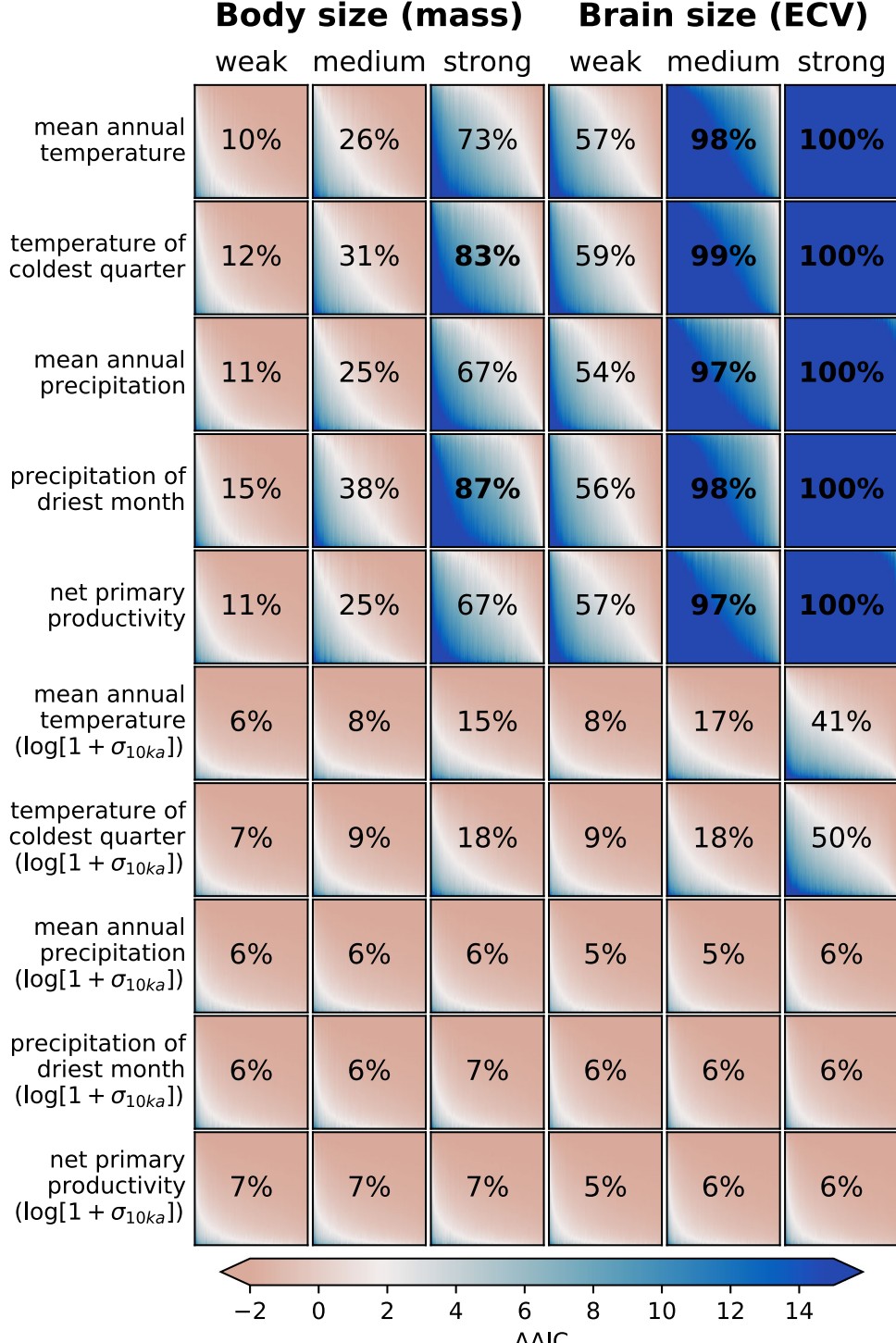

**Fig. 2 Power analysis for body and brain size data with environmental variables.** The power analysis shows the proportion of synthetic datasets ($n =$ 1000) for which a relationship using the LM-TC model is detectable, i.e., $\Delta AIC \geq 2$ (LM-T relative to LM-TC). The color gradient in each panel indicates how many such relationships can be detected within each single synthetic dataset, resolved as vertical bands in each panel. Body and brain size are in log-transformed units. The color bar is chosen so that white reflects the $\Delta AIC$ threshold of 2. Source data are provided as a Source data file.

For brain size, we found a relationship with long-term rainfall variability (MAPvar10), with LM-TC and LM-T*C outperforming LM-T in 78% of thinned datasets (Table 3 and Fig. 3d). In 70% of the cases, LM-TC was the better model, and brain size was found to decrease with increasing levels of long-term rainfall variability (−2.7% per MAPvar10 unit). This effect was the same order of magnitude as the strong effect hypothesized in the power analysis (+/−3.6% per MAPvar10 unit), which had suggested we

had little power to find an effect. In contrast to the results for body size, the additional brain size variance explained by MAPvar10 is only 5%, an order of magnitude smaller compared to variance explained by the differences among taxa (47%). We also found an effect of NPP detected in 62% of thinned datasets by LM-TC and LM-T*C combined. In this case, however, LM-T*C was the most supported model (in 47% of all cases) and the effect of NPP is only different from zero for Mid-Pleistocene

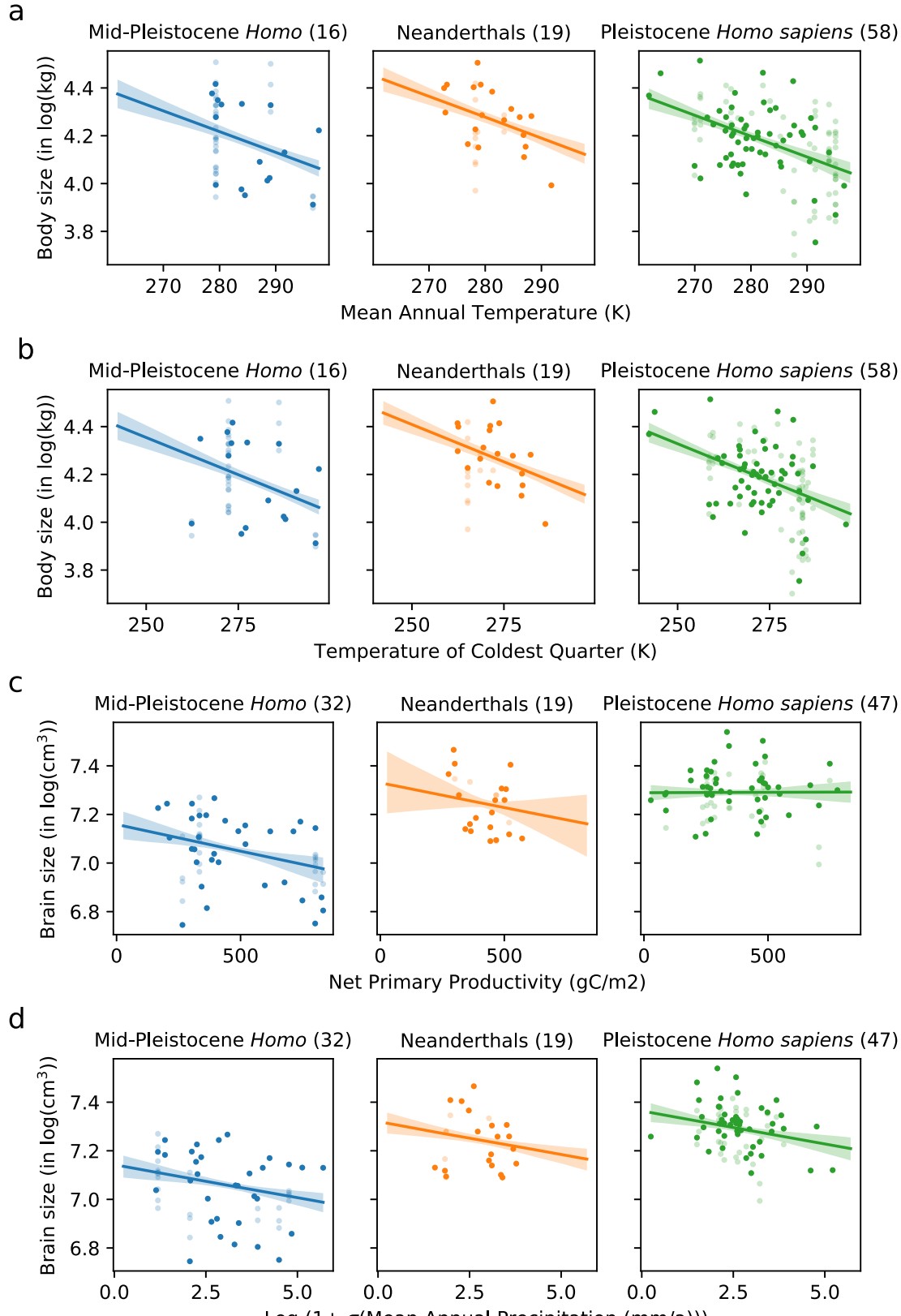

**Fig. 3 Environmental variables associated with body and brain size variation in later *Homo*.** The relationships of environmental variables showing the strongest statistical support for body size among the fossil data are **a** mean annual temperature (LM-TC) and **b** temperature of the coldest quarter (LM-TC); for brain size, these are: **c** net primary productivity (LM-T*C) and **d** long-term variability of annual precipitation (LM-TC). The shaded band corresponds to the 95 percentile range (2.5–97.5%) of all linear regression lines that have been calculated for the 1000 randomized and thinned samples with the thick line in the center corresponding to the median (50 percentile). Each semi-transparent point represents a single fossil record, whereas the opaque points represent a record from a randomly thinned sub-sample. Numbers in brackets indicate the number of fossils used in thinned sub-samples. See Tables 2 and 3 for the size of these effects. Source data are provided as a Source data file.

**Table 2 Analyses of fossil data for body size (log-transformed) and environmental variables in *Homo* based on 1000 randomly thinned datasets.**

| Variable/effect | LM-TC | LM-T*C: MP *Homo* | LM-T*C: Neanderthals | LM-T*C: *Homo sapiens* | R² LM-TC | R² LM-T*C |
|---|---|---|---|---|---|---|
| **MAT** | **−0.87 (−1.1, −0.63)** | −1.0 (−1.6, −0.48) | −1.2 (−1.6, −0.7) | −0.79 (−1.1, −0.5) | **0.21 (0.14, 0.29) [99%]** | 0.22 (0.15, 0.29) [0%] |
| MAP | −0.003 (−0.01, 0.0033) | 0.00065 (−0.012, 0.019) | 0.012 (0.0012, 0.023) | 0.0082 (−0.015, −0.00045) | 0.06 (0.03, 0.09) [0%] | 0.09 (0.04, 0.14) [4%] |
| **Mean temperature of coldest quarter** | **−0.62 (−0.76, −0.48)** | −0.49 (−0.71, −0.28) | −1.0 (−1.4, −0.63) | −0.61 (−0.8, −0.41) | **0.20 (0.14, 0.26) [99%]** | 0.21 (0.14, 0.27) [0%] |
| Mean precipitation of driest quarter | 0.0062 (0.0014, 0.011) | 0.019 (−0.0044, 0.039) | 0.016 (0.0097, 0.022) | −7e−05 (−0.0067, 0.0069) | 0.06 (0.03, 0.10) [4%] | 0.09 (0.05, 0.15) [3%] |
| NPP | −0.015 (−0.025, −0.0036) | −0.0091 (−0.033, 0.02) | 0.0084 (−0.017, 0.038) | −0.022 (−0.035, −0.0079) | 0.08 (0.04, 0.14) [30%] | 0.10 (0.05, 0.17) [11%] |
| MATvar10 | 4.3 (−2.6, 11.0) | 24.0 (−9.4, 59.0) | 7.7 (−18.0, 32.0) | 0.81 (−6.3, 7.5) | 0.06 (0.03, 0.10) [1%] | 0.09 (0.05, 0.17) [14%] |
| MAPvar10 | −1.9 (−3.8, 0.47) | −4.7 (−11.0, 1.6) | 1.4 (−2.7, 6.2) | −1.8 (−4.5, 1.2) | 0.07 (0.03, 0.12) [8%] | 0.09 (0.04, 0.17) [11%] |
| Mean temperature of coldest quarter (var10) | 4.8 (−2.4, 13.0) | 20.0 (−4.1, 44.0) | 6.7 (−11.0, 23.0) | −0.097 (−8.5, 8.8) | 0.06 (0.03, 0.11) [3%] | 0.09 (0.04, 0.16) [11%] |
| Mean precipitation of driest quarter (var10) | 0.48 (−0.64, 1.7) | 1.7 (−1.7, 6.0) | 2.5 (0.67, 4.4) | −0.59 (−2.1, 0.99) | 0.05 (0.03, 0.09) [0%] | 0.08 (0.04, 0.13) [2%] |
| NPPvar10 | 0.7 (−1.3, 2.7) | −2.5 (−10.0, 5.1) | 1.8 (−2.3, 6.1) | 1.2 (−0.93, 3.6) | 0.05 (0.03, 0.09) [1%] | 0.07 (0.04, 0.14) [2%] |

The median effect sizes (slope) of the LM-TC and LM-T*C models (in percentage change per climate variable unit) are shown with their 95% range (2.5–97.5%) based on the 1000 thinned datasets. MAT and mean temperature of coldest quarter are highlighted in bold because LM-TC is the best model based on the difference in the AIC (ΔAIC ≥ 2). No highlight for a given variable means that LM-T is the best model among all three. The percentage in [] indicates how often (out of the 1000 randomly thinned datasets) a given model was the best among all models. The $R^2$ values (with their 95% range) indicate how much more variance in body size can be explained compared to the LM-T model. For reference, the $R^2$ of the null model LM-T is 0.05 (0.03, 0.08).

**Table 3 Analyses of fossil data for brain size (log-transformed) and environmental variables in *Homo* based on 1000 randomly thinned versions.**

| Variable/effect | LM-TC | LM-T*C: MP *Homo* | LM-T*C: Neanderthals | LM-T*C: *Homo sapiens* | R² LM-TC | R² LM-T*C |
|---|---|---|---|---|---|---|
| MAT | −0.15 (−0.27, −0.046) | −0.48 (−0.77, −0.21) | 0.072 (−0.43, 0.58) | −0.046 (−0.17, 0.082) | 0.48 (0.44, 0.51) [0%] | 0.49 (0.45, 0.53) [5%] |
| MAP | −0.0056 (−0.01, −0.0015) | −0.01 (−0.018, −0.0041) | −0.011 (−0.031, 0.0029) | 0.0012 (−0.0039, 0.0066) | 0.49 (0.45, 0.53) [13%] | 0.51 (0.46, 0.56) [29%] |
| Mean temperature of coldest quarter | −0.093 (−0.17, −0.031) | −0.24 (−0.41, −0.098) | −0.24 (−0.54, 0.11) | −0.0075 (−0.079, 0.068) | 0.48 (0.44, 0.51) [0%] | 0.49 (0.45, 0.53) [0%] |
| Mean temperature of coldest quarter | −0.0015 (−0.0059, 0.0025) | −0.0012 (−0.012, 0.0081) | −0.0017 (−0.0078, 0.0041) | −0.0015 (−0.0081, 0.0049) | 0.47 (0.44, 0.51) [0%] | 0.48 (0.44, 0.51) [0%] |
| **NPP** | −0.011 (−0.019, −0.0041) | **0.022 (−0.036, −0.0097)** | −0.02 (−0.053, 0.0096) | 0.00023 (−0.0086, 0.0091) | 0.49 (0.45, 0.53) [15%] | **0.52 (0.46, 0.58) [47%]** |
| MATvar10 | 2.0 (−3.7, 8.4) | 14.0 (0.94, 30.0) | −6.1 (−21.0, 8.4) | −1.7 (−8.5, 4.4) | 0.47 (0.44, 0.51) [0%] | 0.50 (0.45, 0.56) [25%] |
| **MAPvar10** | **−2.7 (−4.2, −1.2)** | −3.3 (−6.3, −0.81) | −3.3 (−9.0, 1.8) | −2.0 (−3.7, −0.23) | **0.51 (0.46, 0.56) [70%]** | 0.52 (0.46, 0.58) [8%] |
| Mean temperature of coldest quarter (var10) | −0.46 (−5.6, 5.0) | 8.8 (−1.3, 21.0) | −7.1 (−18.0, 2.7) | −6.3 (−13.0, 0.55) | 0.47 (0.44, 0.51) [0%] | 0.50 (0.45, 0.55) [22%] |
| Mean precipitation of driest quarter (var10) | −1.3 (−2.2, −0.24) | −2.7 (−4.7, −0.58) | −2.0 (−3.8, −0.046) | 0.23 (−1.1, 1.5) | 0.48 (0.44, 0.52) [10%] | 0.50 (0.45, 0.55) [15%] |
| NPPvar10 | −1.1 (−2.7, 0.41) | −1.2 (−4.7, 1.9) | −3.0 (−7.6, 1.3) | −0.52 (−2.1, 1.2) | 0.48 (0.44, 0.52) [6%] | 0.49 (0.44, 0.53) [3%] |

The median effect sizes (slope) of the LM-TC and LM-T*C models (in percentage change per climate variable unit) are shown with their 95% range (2.5–97.5%) based on the 1000 thinned datasets. MAPvar10 is highlighted in bold because LM-TC is the better model than LM-T based on the difference in the AIC (ΔAIC ≥ 2). NPP is highlighted in bold because LM-T*C is better than the null model, LM-T. No highlight for a given variable means LM-T is the best model. The percentage in [] means that LM-T is the best model among all three. The percentage in [] indicates how often (out of the 1000 randomly thinned datasets) a given model was the best most among all three. The $R^2$ values (with their 95% range) indicate how much more variance in brain size can be explained compared to the LM-T model. For reference, the $R^2$ of the null model LM-T is 0.47 (0.43, 0.51).

*Homo* [−0.022%/(gC/m²)] (Fig. 3c and Table 3). If we were to consider Mid-Pleistocene *Homo* only, the explained variance by NPP would amount to 15% (95% range: 4–31%) with an effect size of −0.022%/(gC/m²). No other tested variable was a good predictor of brain size (Table 3 and Supplementary Figs. 6–9).

## Discussion

Climatic fluctuations and ecological factors have frequently been proposed as potential drivers of brain and body size evolution within the hominin lineage[8–12,23]. This study presents the first systematic attempt to quantitatively test different environmental effects on body and brain size variation for the genus *Homo* during the past ~1 Ma. The climate variables we investigated are representative of the climatological mean (30-year averages) for each 1000-year period of the past ~1 Ma. Hence, climate oscillations on sub-millennial time scales, which might have had some impact on human body and brain size evolution, are not resolved, but such a finer resolution is also precluded by the inherently larger dating uncertainty of Pleistocene human fossils.

We found that MAT is uniformly associated with body size across Mid-Pleistocene *Homo*, Neanderthals, and Pleistocene *H. sapiens*. The extent of this relationship is greater than that estimated for modern humans in a recent study[31]. The direction of this association supports some of the predictions of the Environmental Stress Hypothesis, with temperature (i.e., thermal stress) being the key driver: larger body sizes are consistently found in colder regions, where both annual mean and mean coldest quarter temperature are lower. These findings fit the general expectations of Bergmann's rule and are consistent with some—though not all[33,38]—previous studies on humans, hominins, and other animals[8,10,22,31]. Following this interpretation, short-term challenges resulting from colder temperature experienced by hominin populations (thermal stress) were apparently countered via phenotypic adaptation toward larger bodies as a buffer mechanism, either through natural selection, plasticity, or a combination of both. We failed to detect any effect of low rainfall or nutrient-poor environments as determinants of stress in our analyses.

Our analyses detected no such association of temperature with brain size. We did find relationships with the 10 ka-sigma of mean annual precipitation (MAP) and NPP, but the variance in brain size explained by these variables was small compared to the effect of MAT on body size. These results suggest that brain size within *Homo* is less influenced by environmental variables than body size during the past 1.0 Ma. Apart from other drivers being likely more relevant (see below), one factor contributing to the difficulty of detecting environmental effects lies in the strong performance of the null model (LM-T) based on taxonomic differences in brain size variations that explained much more variance ($R^2 = 0.47$) compared to body size ($R^2 = 0.05$). This being said, our analyses suggest that brain sizes tend to be higher in regions of low NPP and smaller in more productive regions, although this only holds for Mid-Pleistocene *Homo* but not for Neanderthals or Pleistocene *H. sapiens*. This negative correlation is not necessarily a direct effect of environments on human phenotype but can rather be interpreted as an indirect interaction of behavioral changes with environmental variables: regions with lower NPP feature more open steppe and grassland habitats with more frequent large mammals and particularly bovids ("productivity paradox"; ref. 39). As such, our findings can be related to changes in subsistence strategies toward more frequent and systematic hunting of larger-sized bovids in these environments, in association with cognitive changes toward more complex weapons and coordinated group activity. The lithic, faunal, and isotopic records show an increase of such behaviors and ecosystems inhabited by *Homo* throughout the Middle Pleistocene that

supports this interpretation[40–43]. The divergent pattern in Neanderthals and Pleistocene *H. sapiens* might be due to an already higher established brain size close to the physiological maximum during colonization of more northern latitudes (>40°; *H. sapiens*: mean = 1505 cm³, $n = 37$; Neanderthals: mean = 1398 cm³; $n = 25$), while the other taxon either evolved in situ in these areas or had higher growth potential. More early African *H. sapiens* fossils are required to adequately test this interpretation.

Our fossil data show a relationship between long-term variation in rainfall (MAPvar10) and brain size that is of opposite sign than expected from the Environmental Variability Hypothesis[11,36]. Instead, this prediction is consistent with the Environmental Consistency Hypothesis: larger brain sizes occur in more stable environments across all studied *Homo* taxa. This result is likely an effect of brain growth being constrained by reduced resource availability and predictability over multi-millennial scales, acting as an extinction filter.

Our linear models did not find associations with 10-ka variability measures for other environmental variables in either body or brain size. We also failed to find support for the Environmental Constraints Hypothesis (Table 1). However, we need to be careful in interpreting these negative results. The fossil hominin record is scarce and patchy in space and time, confounding the ability to find patterns in our data[26]. We thus modeled and analyzed synthetic datasets to assess the degree to which the intrinsic nature of the fossil record biases and distorts associations of body and brain size with environmental variables. The power analysis shows that we should have been able to detect at least medium to strong associations between brain size and MAT, MAP, mean temperature of the coldest quarter, and mean precipitation of the driest quarter (Fig. 2). The synthetic data thus suggest that our negative results for these variables, and the lack of support for the Environmental Stress and the Environmental Constraints Hypothesis, are either "true negative" findings or that true effect sizes are relatively small. On the other hand, we had little power to detect associations of body and brain size with long-term climate variability (i.e., the consequences of the Environmental Variability and the Environmental Consistency Hypotheses), leaving them as potential targets for future analyses with even larger sample sizes.

There are several implications from our study for human evolution that point toward future analyses. Many standard models and recent accounts of the origins, bio-cultural evolution, and dispersal of our genus and species have invoked environmental drivers as prime movers[9,44–46]. Yet, necessary temporal correlations of paleoanthropological and archeological data with environmental information have been plagued by issues of resolution, scale, and data availability[47]. Using emulated global climate model data[27], this study shows that different climatic variables predict human brain and body size evolution over the past 1 Ma. These findings have implications beyond human evolution. The scaling between body size and brain size is remarkably consistent across vertebrates, but increased variability in brain growth appears to underpin observed patterns of encephalization among birds and mammals[48]. Consistency in the observed patterns of encephalization within lineages is often attributed to developmental constraints that link the ontogenetic trajectories of brain and body size, although there is emerging evidence that deviations from the patterns found in mammals and primates may be driven by functional variation and different selective pressures[49]. Such adaptive mechanisms likely underpin the variation in brain development observed in Pleistocene hominins[50]. The demonstration that brain and body size evolution were influenced by different environmental factors supports this broader interpretation of unique selective pressures driving phenotypic diversification in the hominin lineage.

We also note that many of the environmental variables provided no detectable correlations and explained variance is often low, raising doubts about an unquestioned a priori reliance on environmental factors in explaining macro-processes in human evolution. There is a need for more quantitative tests of such hypotheses in explicitly formulated theoretical frameworks. Future work on these questions could (i) expand analyses on environmental drivers into the entire Pleistocene and Pliocene and (ii) examine other proposed drivers that are not tested here (see below). There are ample changes in the size of endocranial volumes and body mass between ~5 and 1 Ma among taxa of *Ardipithecus*, *Australopithecus Paranthropus*, and *Homo* that could be the result of climatic forcing and ecological adaptations, or yet other factors. This period also constitutes the focus of the original variability hypothesis[11,36]; however, the fossil record ~5–1 Ma has lower sample sizes per taxon and is patchier in time and space. While we gathered datasets for body and brain size back to 4.4 Ma[6], we refrained from extending our analyses to this period as the current quality of the fossil data with the added uncertainty of climate models >1 Ma renders such studies more speculative. Improved paleoclimate models and new discoveries with good chronometric ages and taxonomic information will eventually allow for such studies.

In the meantime, testing other proposed drivers of human body and particularly brain size could be more fruitful. Interspecies competition and niche exclusion likely drove some of the observed significant differences in brain and body size between (sympatric) species of *Homo* (e.g., refs. [2,4,18]), including shifts to larger social groups or communication networks driving further encephalization[16]. Archeologically established changes in subsistence patterns likely played a role as the nutritional basis allowing for the evolution of larger bodies and the maintenance of energetically costly brains[14,15,51,52], and we have found indirect evidence to support this in our study. Yet the spatio-temporal trajectories and taxonomic associations of these behaviors in the archeological record are not well resolved. Finally, there is a long-standing debate about a feedback process between culture, cognition, and encephalization. Increased reliance on technology and material culture might have started a long-term directed evolutionary process selecting for advanced cognition and larger brains[19–21], with greater detachment from direct environmental effects, particularly in *H. sapiens*. In parallel with brain size increases, stone tool technology showed major changes over the past 2 million years[53] with an accelerated pace of cultural change by ~300 ka and again with the onset of the Upper Paleolithic and Later Stone Age[17,54–56].

While many of these factors might have played a key role in body and/or brain size evolution, future models should include interacting components[57,58] such as the co-evolution of changing environments, subsistence, and technology in driving brain evolution[14,18,51,52,59]. Such potential influences on hominin brain and body size need to be tested by formulating and testing explicit hypotheses with statistical analyses. This strategy requires innovative ways to translate the often qualitative archeological information into comparable quantitative data, potentially via machine learning methods. In this study, the support or falsification of certain environmental hypotheses to explain body and brain size changes among *Homo* in the past million years exemplify the usefulness of this approach.

## Methods

**Body and brain size database**. The fossil dataset consists of the hitherto largest collection of body ($n = 204$) and brain size estimates ($n = 166$) from *Homo* in the past ~1.0 Ma (Fig. 1). The data on hominin body size estimates are derived from our own previous study[6] plus additional estimates[60] and updated chronometric ages from more recent literature. Individual body size estimates are provided by

specimen in Supplementary Data 1 with data sources. The bulk of data on hominin brain sizes (endocranial volume, in cm³) is derived from recent meta-analyses[7,12,13,61,62] and updated chronometric information. Specific sources of these data are indicated in Supplementary Data 2, with some assessments bearing larger errors due to the incomplete state of the crania on which they are based (e.g., Arago 21, Vértesszőlős, Zuttiyeh). Each body and brain size estimate is associated with information on estimated chronometric age (dating method and data source), geographical location (longitude and latitude), and taxonomic attribution. For the exact locations per specimen, see interactive map in Supplementary Note 1. We divided the dataset into three taxonomic units: Pleistocene *H. sapiens*, Neanderthals, and Mid-Pleistocene *Homo*. Whereas hypodigms of *H. sapiens* and Neanderthal remains are generally agreed upon, we use "Mid-Pleistocene *Homo*" as a strictly analytical unit to denote African and European Middle Pleistocene hominins that predate Neanderthals and are not assigned to *Homo naledi*, between ~800 and 130 ka. We refrained from further division of this group due to the often fragmentary nature of fossils, unclear alpha taxonomy, and small sample size. Analyses performed within these taxonomic units minimize phylogenetic effects of, e.g., significantly different brain sizes (e.g., ref. [2]). Specimens from *H. naledi* and *Homo floresiensis* had to be excluded from this analysis as for each taxon they derive from a single location and age bracket, precluding assessment of paleoclimatic variation. Limitations to the fossil datasets (see e.g., refs. [2–4,6]) include imprecision of brain and body size estimates due to methodical and taphonomic problems, uncertainties of absolute ages that translate into uncertainties of the associated climate, and unequal sampling of hominin fossils across time and space. These limitations were incorporated into the construction of the synthetic dataset to assess the extent of their effects on the overall results for the actual fossil dataset. For all further analyses, brain and body size values were log-transformed as they increase multiplicatively.

**Climate reconstructions**. Each body and brain size estimate required corresponding estimates of relevant climatic variables. Our climate records are numerical model estimates based on global climate reconstructions for the past 1 Ma using the global climate model emulator GCMET[27]. The main idea behind GCMET is that global climate model (GCM) simulations of the past 120,000 years contain sufficient information about long-term climatic changes on time scales of ≥1000 years. Given that we know the external boundary conditions, we can reconstruct previous glacial–interglacial climatic changes. The Quaternary climate is largely determined by dynamics of the Northern Hemisphere ice sheets, which, in turn, are affected by orbital variations of the Earth around the Sun and variations of atmospheric $CO_2$. Using these factors as external boundary conditions, GCMET can emulate the climate of the Quaternary in a similar way as a state-of-the-art GCM[27].

The atmospheric $CO_2$ record of the past 1 Ma that we use in this study is a composite of the EPICA $CO_2$ record from an Antarctic ice core[63] and of output from a carbon cycle model (CYCLOPS)[64]. The EPICA record covers the past ~800 ka, whereas we use the CYCLOPS model output to cover the time up until 1.0 Ma. Orbital variations are based on calculations by Berger and Loutre[65]. Ice-sheet extents for the past 800 ka are based on numerical ice-sheet model output[66]. For the period before 800 ka, we assumed present-day ice-sheet configurations. This is an appropriate assumption given that all but one specimen of the fossil record before 800 ka in our datasets are within Africa or southeast Asia and thus far away from ice-sheet margins, with the local GCMET climate reconstructions not affected by this simplification.

For each fossil site location from the body and brain size database, we extracted a time series of the relevant climate variables, see Table 1 (also Supplementary Figs. 10 and 11). The time series were used to attach the value of each climate variable to the fossil record, both for the actual fossil data as well as for the synthetic fossil datasets.

**Linear models**. The null and two alternative linear models used throughout this manuscript are defined as follows. The null model simply estimates the mean for each taxonomic group, and we refer to this model as LM-T (linear model with taxa):

$$Y = \beta_0 + \beta_1 \times \text{taxon} \tag{1}$$

Here $Y$ corresponds to either body or brain size (or the log-transformed thereof), whereas $\beta_0$ is the intercept, which is equivalent to the mean size of the reference taxon, and $\beta_1$ is a factor that reflects the deviation from this mean size for a taxonomic group (thus giving independent intercepts for Mid-Pleistocene *Homo*, Neanderthals, or Pleistocene *H. sapiens*).

The first alternative model contains the effect of the climate variable $X$ (across all taxa):

$$Y = \beta_0 + \beta_1 \times \text{taxon} + \beta_2 \times X \tag{2}$$

Here $\beta_0$ and $\beta_1$ are the intercept terms, giving taxon-specific values, and $\beta_2$ is the slope, which is the same across all taxa. We refer to this model as LM-TC (linear model with taxonomic differences plus a climate effect).

The second alternative model takes taxonomic differences for the slope of the climate effect into account. This is done via an interaction term, $\beta_3$, which acts as a modifier for the slopes (i.e., different intercepts, given by $\beta_0$ and $\beta_1$, and slopes,

given by $\beta_2$ and $\beta_3$, for each taxonomic group):

$$Y = \beta_0 + \beta_1 \times \text{taxon} + \beta_2 \times X + \beta_3 \times \text{taxon} \times X \qquad (3)$$

We refer to this model as LM-T*C (linear model with taxonomic differences plus a taxon-specific climate effect). The slopes, $\beta_2$ and $\beta_3$ in Eqs. (2) and (3), respectively, are presented in the main text in Tables 2 and 3 (for the log-transformed sizes) and in Supplementary Tables 1 and 2 (for the natural units of the sizes).

**Synthetic datasets and power analysis.** Apart from determining the smallest sample size suitable to detect the effect of a given test at the desired level of significance, power analysis can also be used as a formal way to test whether a relationship between dependent and independent variables can be detected with the available data and proposed methods (i.e., linear models in our case) assuming that such a relationship exists. Before testing for any true association between local climate and the fossil record, we use such a power analysis to assess our power to detect relationships of different effect sizes given the uncertainties, for example, in body/brain sizes, dating, and climate reconstructions. We generated 1000 synthetic datasets for each of the ten climate variable associations (MAP, MAT, NPP, mean temperature of coldest quarter, mean precipitation of driest quarter, and the logarithm of their running standard deviation over a 10,000-year window) with body and brain size. For each association, we assumed a *strong*, a *medium*, and a *weak* relationship between size and climate.

By *strong*, we refer to 1/4 of the maximum possible slope given by the range of the climate and size. Subsequently, *medium* is half the slope of the *strong* relationship, (1/8 maximum possible slope), and *weak* is half the slope of the *medium* relationship (1/16 maximum possible slope). For example, the *strong* association between MAT and body size (Bergmann's rule) is $-0.34 \, \text{kg/°C}$, based on the above defined rule. This is close to the estimated association between temperature and body size of about $-0.4 \, \text{kg/°C}$ found for modern humans in a recent study (ref. [31], their Fig. 5A). Unfortunately, there are no empirical data about other climatic relationships and body (or brain) size. For simplicity, we therefore applied the same rule of *strong*, *medium*, and *weak* associations for all other climate variables and for brain size.

Before generating a synthetic dataset, we estimated the intercepts $\beta_0$ and $\beta_1$ and the slope $\beta_2$ for the LM-TC model, Eq. (2). However, for the real fossil analysis we used the model with the interaction term, LM-T*C, Eq. (3). First, we looked up the climate record for each location and time from the climate time series and attached it to the respective empirical fossil records. We calculate the maximum slope from the $X$ and $Y$ ranges as $\beta_1 = \text{range}(Y)/\text{range}(X)$. Assigning an actual relationship factor, e.g., *strong* (=1/4), the intercept $\beta_0$ can be calculated using the $X$- and $Y$-midpoints, $\beta_0 = Y_{\text{midpoint}} - 1/4\beta_1 X_{\text{midpoint}}$.

For the synthetic fossil datasets, we assume an age uncertainty range of 10% (±5%) for radiocarbon-dated fossils, i.e., younger than 50 ka cal BP (e.g. ref. [67]), and 20% (±10%) for fossils older than 50 ka coming from other dating methods with higher uncertainty such as luminescence, U-series, or ESR (e.g., ref. [68]). Furthermore, we assume a standard error of 2 K for mean annual and mean temperature of coldest quarter. For all other climate variables, we assume a 20% error range (±10%). The 2 K and the 20% are in line with climate model biases as estimated in a recent study[69]. Within a taxonomic unit of the genus *Homo*, we assume a coefficient of variation (CV) of 7% for body size (average of intrapopulation means of 19 global Holocene hunter-gatherer populations, $n = 510$, data from JTS) and 3.5% for brain size (from ref. [28] populations, dataset: http://volweb.utk.edu/~auerbach/HOWL.htm; ref. [70], see also ref. [32]). Previous research has demonstrated that the range of body size variation in Holocene human populations is larger than any taxonomic unit of earlier hominin and encompasses the range of variation found within earlier hominins[6] and that sexual dimorphism in size among Mid-Pleistocene hominins is comparable to that of modern humans[71]. While there are significant differences in brain shape through recent hominin evolution, the range of size variation within Pleistocene hominin taxa remains comparable to that observed among modern humans[60]. These observations suggest that modeling the intrapopulation variation among hominin taxa upon modern human coefficients of variation provides a reasonable estimate of variation within hominin taxa that are often presented only by much smaller sample sizes. To create a synthetic dataset that has a mean and a variance as close to the fossil dataset, we introduced taxonomic size differences ($\beta_1$ in Eq. (2)) that is based on the taxonomic differences in the mean size. This difference was estimated directly from the fossil dataset.

The procedure to generate a single synthetic dataset is as follows. First, we selected a relationship strength, e.g., *strong*, and calculated the slope and intercepts. For each synthetic data point, we:

1. Looked up the age and added a randomly sampled error (±5% or ±10%).
2. Looked up the fossil site and selected the climate record from the previously calculated time series for that location and sampled age.
3. Added a randomly sampled error (i.e., S.D. of 2 K or 20%) to the climate record. This is now the $X$ value.
4. Multiplied $X$ with the slope $\beta_2$ and added the intercept $\beta_1$ with the respective taxonomic correction. This translates the climate record $X$ into a size estimate $Y$.

5. Added a random term to $Y$ based on the CV, i.e., 3.5% for brain and 7% for body size.
6. Repeated steps (1)–(5) for each fossil record and saved all locations, ages, $X$s, and $Y$s to a file. This is a single synthetic dataset in the same format as the original fossil dataset.

We repeated this $N$ times to generate $N$ synthetic datasets and repeated the same procedure for the other relationship strengths, i.e., *medium*, and *weak* and for all other climate variables. Panels of exemplary synthetic datasets for body and brain sizes in comparison with the original data are shown in Supplementary Figs. 10 and 11.

We use the same thinning approach as described in the main text ($n = 1000$) for the synthetic datasets. These are then used for a power analysis to test whether the linear relationship between any climate variable and body or brain size can be detected. We fitted both the LM-TC model, Eq. (2) (in which the slope defining the relationship between climate and size is the same for the three taxonomic groups, which can differ in their intercept), and the LM-T*C model, Eq. (3) (different slopes and intercepts for the three groups). A climate effect was deemed present if the null model had a higher AIC value compared to either of the alternative models, LM-TC or LM-T*C, ($\Delta \text{AIC} > 2$), i.e., LM-T, Eq. (1), in which the three groups differ in size but there is no effect of climate. Figure 2 in the main text shows the power to detect a true relationship between size and climate. Individual records are color-coded according to the AIC difference between the LM-T and the alternative models, LM-TC or LM-T*C, ranging from $-2$ (red) to $+15$ (blue) with 2 as midpoint (white).

All statistical tests were undertaken in Python version 3.8.5 using the following Python packages: statsmodels 0.12 (for linear models), pandas 1.1.3 (for dataframes, reading/writing CSV/Excel files), netCDF4 1.5.3 (reading NetCDF files), matplotlib 3.3.2 (for plotting), and numpy 1.19.2 (numerics).

**Reporting summary.** Further information on research design is available in the Nature Research Reporting Summary linked to this article.

## Data availability
All data generated or analyzed during this study are included in this published article and its supplementary information files. Data on fossil specimens (body and brain sizes) and all sources for each data point are provided in Supplementary Data 1 and 2. All results and the climate data to run all analyses in this paper can be accessed via https://doi.org/10.17605/OSF.IO/SMYAC. Information on modern variation of brain size derives from the William W. Howells Craniometric Data Set[70] accessible via http://volweb.utk.edu/~auerbach/HOWL.htm. Source data are provided with this paper.

## Code availability
The source code to run all analyses in this paper can be accessed via https://doi.org/10.17605/OSF.IO/SMYAC.

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

## Acknowledgements

We thank C. Sommer and H. Pehnert for creating the interactive webmap of fossil specimens. J. Beier gave access to her literature database of (Upper) Paleolithic human remains that helped greatly in assigning chronometric ages for all fossils included in this study. This study was supported by a European Research Council (ERC) Consolidator Grant to AM (Local Adaptation, grant nr. 647787), an ERC Consolidator Grant to JTS (ADaPt, grant nr. 617627) and by funding of the Antarctic Science Platform to MK (ANTA1801). We acknowledge support by the Open Access Publishing Fund of University of Tübingen and by Projekt DEAL.

## Author contributions

All authors contributed to the conception and design of the work; M.W., M.K., and J.T.S. acquired data; M.K. and A.M. performed the analysis of data; all authors contributed to interpretation of data; M.W. drafted the manuscript; all authors contributed to revision of the manuscript.

## Competing interests

The authors declare no competing interests.
