## [Peer Review File · Nature Communications]

Reviewers' Comments:

Reviewer #1:

Remarks to the Author:

I found this to be a valuable and overall convincing test of models concerning possible environmental influences on body and brain size evolution, particularly for the larger data sets available in the later part of the record. I have added just a few very minor comments to the attached files but my main concerns are with two aspects of the data sets.

While there are clear indications of data sources for the body and ECV determinations, I could not see clear indications for dating of fossils (unless I missed them) in this submission or the referenced 2017 paper by some of the same authors. Chronological control is very important for a study like this and while some sites are well dated, others have only single determinations that may or may not closely relate to the fossils concerned, conflicts of chronology, or worse still just 'guesstimates' (e.g. Elandsfontein, Ndotu, Eliye Springs, the Gibraltar Neanderthals). So I'd like clear citations of sources for the age estimates used, and discussion of why particular dates have been preferred, where there are conflicts or guesstimates. In some cases such as Ngandong, Broken Hill, Saccopastore and Sunghir, there have also been recent publications with new age estimates that should be referenced. Additionally, it looks like some of the radiocarbon ages are calibrated while others are uncalibrated, and for correlation with environmental events, calibrated age estimates are essential. Again, this needs to be clearly shown and discussed.

The second issue I have with the data set is the climatic record used. The global climate model emulator GCMET is the main reference source but to what extent are sub-Milankovitch climatic oscillations captured, which were clearly very important in repeated, rapid, terrestrial environmental changes for at least the latter part of the time period under consideration here? I would like this potential complication to be discussed, at least.

Also I now see that none of my extensive annotations to the data table spreadsheets have copied across so I'll have to summarise some of the points not already made, below:

Is the TD6 individual an adult?

For incomplete fossils such as Arago 21, Vertesszollos, Zuttiyeh, Florisbad, Border Cave, Saccopastore 2, Fontechevade, can you really provide a reliable ECV?

Also some individuals are probably less than 6 years old (e.g. Skhul 1, Devil's Tower) - does that matter?

Combe Capelle a Neanderthal?

Chris Stringer

Reviewer #2:

Remarks to the Author:

The authors provide a cogent and timely reevaluation of a topic that lies at the foundation of human evolution. The paradigm that the brain-body relationship is driven primarily by brain size has resulted in a one-dimensional focus on brain size in previous work. Much of previous work hereby failed to investigate the influence of body size and thereby likely provided an overly simplified view of a complex issue. In this contribution, the authors go a long way in rectifying this undesirable situation. The analyses are thorough and appropriate. The power analysis is especially helpful and adequately buffers the statistical interpretation for the inevitably low statistical sample size for this type of data (empirically, for this type of data, the data set is impressively exhaustive).

The conclusions provide a cogent reflection on the hypotheses that have driven previous research and how the present results move the field forward (which it undoubtedly does).

My only qualm is that there is no contextualization of the broader question outside of the field of human evolution. The principal assumption that has driven previous research is that brains and bodies are constrained to evolve together. Universal scaling laws have been proposed to restrain the independent evolution of brain or body size from their joint allometric relationship.

Basically, what the authors show here is that these universal scaling laws may not restrain the brain-body relationship in hominins as different factors may influence either brain size or body size. This is an important deviation from traditional thinking and has implications beyond human evolution. To broaden the resonance of their findings, the authors could make this more clear.

Reviewer #3:

Please see attached file.

The authors test proposed environmental drivers of brain and body size in *Homo* over the last one million years. They distill previously proposed hypotheses into four main ones, conduct power analyses to see if they can detect an effect if it was actually there, and then run linear models to estimate these effects. They find temperature drove body size, and NPP and MAP variability drove brain size.

While the analyses are interesting and the manuscript is well-written, there are some major issues that should be addressed.

General comments:

1. I would urge the authors to be careful with using the words “causal” and “mechanism” in the manuscript. Linear models do not allow for the inference of causal mechanisms from data and can only infer associations. For example, there might be a third confounding variable driving the independent and dependent variables, such that there is no causal relationship between the two variables.

While predictions are certainly important in the inference of mechanism, the strength of that inference is directly related to the precision of the prediction (e.g., if a certain theory or model predicted that X affected Y with a resulting slope of 2.45 and this was found with the data, this would certainly be strong support for that mechanism). However, the predictions in Table 1 are very imprecise relative to my example: slopes are predicted to be either positive or negative, and there is a 50% change of getting either (i.e., you could have obtained the right direction just by chance). Therefore, I would again soften your language concerning “causal” and “mechanism”.

2. I am a bit confused by why the power analyses are structured the way they are. My impression of power analyses is that their main purpose is to ascertain the sample size needed to obtain a significant result, given an effect size and the variation surrounding that effect size. It looks like sample size is held constant in your power analyses (as this is determined by the fossil record), as is the variation surrounding the effect size (this is determined by your predetermined error values for your variables). Therefore, only the slope is varied using the three levels.

However, it is unclear whether your preselected error values in the variables are actually similar to what is in your dataset, which could lead to an “apples and oranges” comparison when comparing your empirical results to the simulated power analyses (more on this in “Specific Comments”). Moreover, it is not clearly stated what the weak, medium, and strong slope values are for each of your climate variables from the power analyses (this should be presented in Figure 2, with confidence intervals quantifying the variability seen across your simulations). Therefore, how are we to determine whether the slopes estimated from the data are actually weak, medium, or strong? This is important because the whole point of your power analyses is to determine if we can detect an effect, given that its slope is strong, medium, or weak, so it is important that we know whether the slopes estimated from data are strong, medium, or weak. In other words, you conduct your power analyses, but you do not clearly connect them with your empirical analyses, thereby leaving the audience “in the dark” as to whether the empirical results are “real” effects. I elaborate on this at specific points in your manuscript (under “Specific Comments”).

In the end, I don't think a power analysis is necessary. A significant relationship between some climate variable and body/brain size is prima facie evidence that you have enough statistical power to detect an effect, despite all the noise in your variables. If you do not have a significant result, then you need a larger sample size or more precisely measured variables, but these are constrained by the available data. A power analysis does not add anything to this.

3. Environmental stress hypothesis: I appreciate the authors' attempts to unify and reduce the number of environment-brain/body size hypotheses in the literature, but I would be wary of lumping Bergmann's Rule into the Environmental Stress Hypothesis: Bergmann's rule has to do with surface area to volume ratios and heat loss, with no necessary recourse to environmental stress.

Furthermore, can't hot environments also be viewed as a high-stress environment, which would select for smaller body masses and taller statures? The way lines 111-116 are written makes it seem like there can only be a positive correlation between larger body sizes and environmental stress (i.e., colder environments), which is undercut by my previous sentence.

Also, can't "higher mobility and reduced vulnerability to predation" (Lines 115-116) apply to any environment of a given temperature and aridity? Why is this exclusive to environments that are "colder, drier, and nutrient-poorer"?

I think these are examples of where lumping multiple hypotheses together results in a loss of nuance and explanations/predictions that are now somewhat confusing.

4. I am a bit confused by your distinction between individual and evolutionary scales (e.g., Table 1, Lines 103-107). Isn't the lifetime scale concerned only with phenotypic plasticity and no evolution whatsoever (i.e., why look at the fossil record if you're interested in this scale)? And can't the evolutionary time scale also involve speciation and/or phenotypic evolution by natural selection (i.e., why only mention extinction)? Furthermore, evolution by natural selection operates by culling individuals with certain traits (i.e., an evolutionary scale pattern caused by processes operating at the individual scale)? Why then such a sharp distinction between these two temporal scales? My point is supported by your mention of "behavior flexibility" (individual scale) and its effect on extinction risk over evolutionary time under "Environmental Variability Hypothesis" (Lines 124-126), though this hypothesis is placed under "Evolutionary Scale" in Table 1.

It seems to me that you and previous hypotheses are ultimately interested in evolutionary scale patterns (prima facie evidence for this lies in the fact that you are analyzing the fossil record). These larger-scale patterns may or may not scale up from individual-scale processes and patterns, but the fact that you look at climate variables with temporal resolutions of at least 1,000 years means you are not addressing individual-scale processes (though I'm not 100% certain that this is the temporal resolution used; this information is not clearly presented in the Methods section). Further supporting this is that you are analyzing multiple species and lineages together over a million years.

5. Brain and body size variables should be log-transformed as these increase multiplicatively, not additively. Also, why are your climate variability variables log-transformed?

6. Is it possible for the authors to upload their climate data? The analyses are not replicable nor completely transparent without these data. Likewise, I assume some coding language was used for the power analyses. If so, the code should be uploaded as well.

Specific comments:

1. Keywords: Shouldn't "brain size" be included here?
2. Line 56: Are these "qualitative and verbal assessments"? It seems to me that latitude, global oxygen isotope curves, etc. are quantitative variables.
3. Line 61-62: I am not sure that this was the main takeaway of Maxwell et al.'s paper. They argued that the hominin record is not sampled well enough to correlate those data with climate patterns. Furthermore, they did not investigate or even comment on hominin body and brain size.
4. Line 73: "old world" should be capitalized.
5. Line 74: Are *H. naledi* and *H. floresiensis* included in your analyses? They don't appear to be according to Figure 1. If they're not included, why were they excluded?
6. Table 1: Why temperature of the *colest* quarter and precipitation of the *driest* quarter instead of warmest and wettest, respectively? This reasoning for this decision should be made explicit for the reader. And why isn't precipitation of the driest quarter analyzed for the Environmental Constraints Hypothesis?
7. Line 151: "additive" and "linear" are redundant.
8. Lines 169-170: A citation is needed here.
9. Line 200: How did you determine "detected an association"? According to the Table 2 caption, "A model is highlighted in bold, if it has the strongest support (no highlight for a given variable means that the null taxonomic model is the most supported)". Did you use AIC to determine this? Was this determined from your power analyses (though the reader cannot connect your empirical results to the power analyses results because you do not present slopes from the latter; see my point under "General Comments")?
10. Lines 202-203: I would be careful with your wording here. A "null" result does not mean same or equal to zero. It just means you do not currently have a large enough sample size and statistical power to detect a difference or non-zero effect.
11. Lines 200-210: According to Figure 2a, these results appear to be driven entirely by Late Pleistocene *H. sapiens*, so I would be careful about generalizing these findings to the other two taxonomic groups (this especially pertains to your Discussion). The slopes may be large for these

other taxa (Table 2), but I bet the R2 are very low. (In fact, all this could be said of 2b, and the results don't appear generalizable to Neanderthals in 2c and 2d).

12. Line 225: I find this result surprising, given that the interaction model increased R2 by 0.04 on average. The percentage in [] is certainly much higher, but again, what this means is difficult to interpret given the ambiguity surrounding what "most supported" means.
13. Lines 232-233: What was the median and 95% range calculated on? The 100 thinned datasets? This should be made clear.
14. Tables 2 & 3: The addition of an interaction term only results in non-negligible increases in model R2. You also did not do power analyses for interaction terms (or at least did not present the results), so it is not possible to interpret your interaction results in that light. Given all this, I would say the interaction terms did not add anything to your models.
15. Tables 2 & 3: I would scale at least your climate variables to be SD=1 (you can scale body and brain size so the slopes are comparable between these two dependent variables). It is difficult to compare the linear model slopes across climate variables, given that the variables are measured in such different units.
16. Lines 263-264: I would not interpret this as a real result given the negligible increase in R2 relative to the null model (Table 3).
17. Lines 291-295: Then why wasn't an effect found? It could be that the effect is not there (as you propose) or instead that the observed effect is weak, or that your chosen error values in the power analyses are not matching what is found in your dataset. The slope itself can tell you whether the effect is non-existent, weak, or strong (the slope is unbiased and will give you the right answer on average even in the face of noisy data, though the standard errors will be higher).
18. Line 353: Should this be cm3?
19. Line 359: Why were they useful? It should be stated here and not left to the reader to find the reason in another publication.
20. Line 426: Shouldn't the intercept be equal to Y minus the slope times X? You have it the other way around.
21. Lines 427-430: Age uncertainties are always published along with estimated ages. Why not use those instead of assuming some arbitrary value?
22. Line 430: Why is temperature given a fixed error of 2K when all other variables get a percentage error? This results in a smaller error for temperature than your other climate variables. I wonder

if this is why the only linear models that have non-negligible R2 are those related to temperature in the body size analyses (Table 2).

23. Lines 430-431: Why these values of error instead of some other number? These numbers are not justified nor are there cited references. Therefore, these numbers seem kind of arbitrary.
24. Lines 431-435: What is a “population of a taxonomic unit of *Homo*”? Did you somehow spatiotemporally circumscribe populations within each species? Furthermore, was the CV calculated across all 19 hunter-gatherer populations? Is this estimate applicable to hominin taxa (potentially multiple species within mid-Pleistocene *Homo*) measured over evolutionary time? Surely, there is more variation in the latter. The same criticism applies to the brain size dataset used. Quickly looking at the brain size dataset from Du et al. (2018), six *H. heidelbergensis* specimens have brain sizes with a CV of 35%.
25. Lines 435-437: This seems kind of “brute force”. I would just include taxonomy as a separate variable and let the model estimate the different intercepts for each taxonomic group, i.e., $Y = \beta_0 + \beta_1 * Taxon + \beta_1 X$.
26. Lines 462-464: Did you create a version of Figure 2 for the interaction models? If you didn't, why not? Based on the linear model results (Tables 2 and 3), I assume none of those percentages comparing the interaction to the additive model would be high (cf. Figure 2).
27. References: genus and species names should be italicized.

References

Du, A., Zipkin, A.M., Hatala, K.G., Renner, E., Baker, J.L., Bianchi, S., Bernal, K.H., Wood, B.A., 2018. Pattern and process in hominin brain size evolution are scale-dependent. *Proceedings of the Royal Society B: Biological Sciences*. 285, 20172738.

Reply to reviewers

We found the critical and constructive comments of the 3 reviewers to be extremely helpful during revision of our manuscript, and appreciate the time and effort that they put into the review process. We agree with the majority of the reviewers' comments and have endeavoured to address each of them in the revised contribution, to the extent that we can. This included modifications to the main datasets, a full re-run of our power and fossil analyses and rewriting several parts of the manuscript. Importantly, the results remained principally the same as before, supporting the robusticity of our study.

In the reply to reviewers below we provide a more detailed description of how the criticisms and suggestions have been addressed point by point. Reviewers comments are in normal font, our responses in *italic*.

Reviewer #1 (Remarks to the Author):

1) While there are clear indications of data sources for the body and ECV determinations, I could not see clear indications for dating of fossils (unless I missed them) in this submission or the referenced 2017 paper by some of the same authors. Chronological control is very important for a study like this and while some sites are well dated, others have only single determinations that may or may not closely relate to the fossils concerned, conflicts of chronology, or worse still just 'guesstimates' (e.g. Elandsfontein, Ndutu, Eliye Springs, the Gibraltar Neanderthals). So I'd like clear citations of sources for the age estimates used, and discussion of why particular dates have been preferred, where there are conflicts or guesstimates. In some cases such as Ngandong, Broken Hill, Saccopastore and Sunghir, there have also been recent publications with new age estimates that should be referenced. Additionally, it looks like some of the radiocarbon ages are calibrated while others are uncalibrated, and for correlation with environmental events, calibrated age estimates are essential. Again, this needs to be clearly shown and discussed.

We fully agree with the reviewer and thank him for this constructive and relevant comment. We added the dating method(s), dating source (literature citation) and additional comments to each fossil specimen as new columns in both the body and brain size database (SOM Files 1 and 2). We also provide additional detail (e.g. all C14 dates are now consistently given as cal BP). Controversial ages are discussed in this document as well as a rationale for why we chose a specific estimate (e.g. Elandsfontein). The update information on dating are also provided in the webmap for all specimens (SOM Figure 1). We also want to mention again here that in our models we assume 10% error for all C14 ages and 20% for all non-C14 ages >50 ka to take into account general uncertainties (i.e. +/-50 ka for fossils at age 500 ka). We consider this reporting and approach as the highest and most transparent standard we could have currently achieved, which still provides meaningful data for environmental correlations and is much higher compared to all recent meta-analyses of body and brain size (with the exception of Du et al. 2018 who also report ages and deviations, but not dating methods or further comments). We also updated each fossil specimen with its most recent age estimate (e.g. Dolní Vestonice, Zhoukoudian UC, Sangiran, Ngandong, Broken Hill, Saccopastore, Sunghir, all since 2017). Based on the changing dates for several of the fossils we also re-ran our entire analyses of the fossil data and updated our results etc. accordingly. Importantly, even though approximately one third of the ages in both the brain and body size database were changed (some up to over +/-100 ka), the results remained consistent with our previous findings, lending further support for the robusticity of this study.

2) The second issue I have with the data set is the climatic record used. The global climate model emulator GCMET is the main reference source but to what extent are sub-Milankovitch climatic oscillations captured, which were clearly very important in repeated, rapid, terrestrial environmental changes for at least the latter part of the time period under consideration here? I would like this potential complication to be discussed, at least.

The long-term glacial-interglacial climatic changes due to the orbital variations, known as Milankovitch cycles, are well covered by the underlying climate model simulations (HadCM3) and therefore, by the derived emulated climate reconstructions (GCMET). However, HadCM3 simulations represent the climatological mean (i.e., 30-year average climate) of a thousand year-long snapshot. For each thousand year period for the last ~1 million years, we have an update of the climatology, but climate variations on shorter time scales, i.e., centennial or millennial-scale variability cannot be reconstructed from the available climate model simulations. However, the dating uncertainty of the (specifically older) fossil record is much larger than that (>1000 years generally), so the time stepping of climate reconstructions is well within this age uncertainty and poses no specific problem for our analysis. We agree with the reviewer, however, that sub-millennial scale climate variations had some effect on the environment within the lifetime of each individual, but the fossil record does not provide the necessary temporal resolution (e.g., having many individuals for a certain, longer period of time). Such variations are therefore beyond the scope of this analysis. We extended the description of the used climate reconstruction (Introduction) and briefly discuss sub-millennial climate variations and issues of resolution as a potential complication of our findings (Discussion).

3) More specific comments by reviewer #1 in the following:

- Is the TD6 individual an adult?
 - For incomplete fossils such as Arago 21, Vertesszollos, Zuttiyeh, Florisbad, Border Cave, Saccopastore 2, Fontechevade, can you really provide a reliable ECV?
 - Also some individuals are probably less than 6 years old (e.g. Skhul 1, Devil's Tower) - does that matter?
 - Combe Capelle a Neanderthal? [actual age much younger!??]
- *TD6 hominin 10 is an adult according to Pablos et al. 2013 (quote p. 610 “talus ATD6-95 is tentatively assigned to Hominin 10 of the TD6 sample, an adult male specimen with which the second metatarsal ATD6-70-107 (already published) is also tentatively associated”)*
 - *We kept these specimens as they feature in both of the largest previously published meta-studies of brain sizes in Holloway et al. 2004 and Bailey & Geary 2009. We noted the fact that some of the ECV estimates are more unreliable, however, in the text and the SOM.*
 - *We removed Skhul 1, Devil's Tower and Engis 2 from the database as they were too young for a reliable ECV estimate.*
 - *Combe Capelle was mistakenly labelled as Neanderthal and is correctly labelled now; the age estimate was also adjusted.*

Based on the removal and/taxonomic re-assessment of some of the fossils – and the updated chronology – we re-ran our entire analyses of the fossil data.

4) I have added just a few very minor comments to the attached files.

All remarks and comments in the marked file were incorporated into the manuscript text.

Reviewer #2 (Remarks to the Author):

1) My only qualm is that there is no contextualization of the broader question outside of the field of human evolution. The principal assumption that has driven previous research is that brains and bodies are constrained to evolve together. Universal scaling laws have been proposed to restrain the independent evolution of brain or body size from their joint allometric relationship.

Basically, what the authors show here is that these universal scaling laws may not restrain the brain-body relationship in hominins as different factors may influence either brain size or body size. This is an important deviation from traditional thinking and has implications beyond human evolution. To broaden the resonance of their findings, the authors could make this more clear.

We thank the review for the praise of the general relevance of our findings and added a new part to our discussion (lines 320-331) contextualizing and broadening the relevance of our findings also outside of the field of human evolution.

Reviewer #3 (Remarks to the Author):

1) I would urge the authors to be careful with using the words “causal” and “mechanism” in the manuscript. Linear models do not allow for the inference of causal mechanisms from data and can only infer associations. For example, there might be a third confounding variable driving the independent and dependent variables, such that there is no causal relationship between the two variables. While predictions are certainly important in the inference of mechanism, the strength of that inference is directly related to the precision of the prediction (e.g., if a certain theory or model predicted that X affected Y with a resulting slope of 2.45 and this was found with the data, this would certainly be strong support for that mechanism). However, the predictions in Table 1 are very imprecise relative to my example: slopes are predicted to be either positive or negative, and there is a 50% change of getting either (i.e., you could have obtained the right direction just by chance). Therefore, I would again soften your language concerning “causal” and “mechanism”.

We agree with the reviewer that one has to be careful with the words “causal” and “mechanisms” in the context of this paper. We want to emphasize that throughout the original manuscript these words were only used in the introductory sections and in contexts where they are frequently used in the general literature (i.e. “causal” only once in the abstract), but never as part of the interpretation of our results. As an example, we intentionally chose “prediction” as the key word in the title of our manuscript. This being said, we slightly changed the phrasing throughout the text in accordance with this comment.

2) I am a bit confused by why the power analyses are structured the way they are. My impression of power analyses is that their main purpose is to ascertain the sample size needed to obtain a significant result, given an effect size and the variation surrounding that effect size. It looks like sample size is held constant in your power analyses (as this is determined by the fossil record), as is the variation surrounding the effect size (this is determined by your predetermined error values for your variables). Therefore, only the slope is varied using the three levels. However, it is unclear whether your preselected error values in the variables are actually similar to what is in your dataset, which could lead to an “apples and oranges” comparison when comparing your empirical results to the simulated power analyses (more on this in “Specific Comments”). Moreover, it is not clearly stated what the weak, medium, and strong slope values are for each of your climate variables from the power analyses (this should be presented in Figure 2, with confidence intervals quantifying the variability seen across your simulations). Therefore, how are we to determine whether the slopes estimated from the data are actually weak, medium, or strong? This is important because the whole point of your power analyses is to determine if we can detect an effect, given that its slope is strong, medium, or weak, so it is important that we know whether the slopes estimated from data are strong, medium, or weak. In other words, you conduct your power analyses, but you do not clearly connect them with your empirical analyses, thereby leaving the audience “in the dark” as to whether the empirical results are “real” effects. I elaborate on this at specific points in your manuscript (under “Specific Comments”). In the end, I don’t think a power analysis is necessary. A significant relationship between some climate variable and body/brain size is prima facie evidence that you have enough statistical power to detect an effect, despite all the noise in your variables. If you do not have a significant result, then you need a larger sample size or more precisely measured variables, but these are constrained by the available data. A power analysis does not add anything to this.

We thank the reviewer for this comment and see that we have not explained our usage of the power analysis in the context of this paper well enough. We now do this in lines 447-456. In short, a power analysis can work in both ways. Either, to ascertain the required sample size given an effect size, or to ascertain the effect size given the sample size. In our case, we do not know the effect size and we are limited by the number of available samples (i.e., fixed sample size). Therefore, we use the power analysis to test how large an effect size has to be in order to be able to detect it, given the fixed sample size. For practical (computational) reasons, we cannot test the effect size gradually, thus we have decided to test for “weak”, “medium”, and “strong” effect sizes. We see that we can also be more explicit about what those different categories actually mean. Our idea for the different effect size categories (“weak”, “medium”, “strong”) rests on the existing literature for the effect size of

Bergmann's rule for humans (Foster & Collard 2013). From this figure, we generalized the effect sizes for the other environmental factors (e.g., net primary productivity, annual mean precipitation, etc) into the three aforementioned ones. As a further reply to the reviewer, we now provide the actual effect sizes (in Supplementary Table 3 and SOM File 3) that we have used for our power analysis and describe our subjective decisions for the three categories more explicitly in the method section (lines 457-465).

3) Environmental stress hypothesis: I appreciate the authors' attempts to unify and reduce the number of environment-brain/body size hypotheses in the literature, but I would be wary of lumping Bergmann's Rule into the Environmental Stress Hypothesis: Bergmann's rule has to do with surface area to volume ratios and heat loss, with no necessary recourse to environmental stress. Furthermore, can't hot environments also be viewed as a high-stress environment, which would select for smaller body masses and taller statures? The way lines 111-116 are written makes it seem like there can only be a positive correlation between larger body sizes and environmental stress (i.e., colder environments), which is undercut by my previous sentence. Also, can't "higher mobility and reduced vulnerability to predation" (Lines 115-116) apply to any environment of a given temperature and aridity? Why is this exclusive to environments that are "colder, drier, and nutrient-poorer"?

I think these are examples of where lumping multiple hypotheses together results in a loss of nuance and explanations/predictions that are now somewhat confusing.

*For this paper we wanted to re-formulate and streamline the range of hypotheses put forward for environmental variables driving brain and body size in Homo (from the literature as cited on page 4 of our manuscript). While the reviewer is correct in pointing out that more combinations of factors (and thus more hypotheses) are possible, here we are interested in a set of clearly formulated hypotheses that can be tested with appropriate environmental variables that are relevant (i.e. have been proposed, see page 4 literature) for human evolution - e.g. Bergmann's rule has frequently been proposed for larger body sizes. We believe that this grouping helps in the bewildering jungle for potential hypotheses and their various and varying names. We also point out that neither reviewer #2 nor #3 had any issue with the grouping of the hypotheses. This being said, we changed our text accordingly to the comment by the reviewer, adding that we look at hypotheses particularly relevant within the context of hominin evolution, though others are conceivable (lines 113-118). We also clearly emphasize Bergmann's rule as a special case of the broader Environmental Stress Hypothesis in the discussion of our results. Since one of the main trends in hominin evolution is **increasing** body and brain size over time (see introduction) this is the main topic of interest (and not smaller body masses). We do not test for stature in the manuscript. We changed our text accordingly to clarify the reasoning behind this decision in lines 113-118.*

4) I am a bit confused by your distinction between individual and evolutionary scales (e.g., Table 1, Lines 103-107). Isn't the lifetime scale concerned only with phenotypic plasticity and no evolution whatsoever (i.e., why look at the fossil record if you're interested in this scale)? And can't the evolutionary time scale also involve speciation and/or phenotypic evolution by natural selection (i.e., why only mention extinction)? Furthermore, evolution by natural selection operates by culling individuals with certain traits (i.e., an evolutionary scale pattern caused by processes operating at the individual scale)? Why then such a sharp distinction between these two temporal scales? My point is supported by your mention of "behavior flexibility" (individual scale) and its effect on extinction risk over evolutionary time under "Environmental Variability Hypothesis" (Lines 124-126), though this hypothesis is placed under "Evolutionary Scale" in Table 1.

It seems to me that you and previous hypotheses are ultimately interested in evolutionary scale patterns (prima facie evidence for this lies in the fact that you are analyzing the fossil record). These larger-scale patterns may or may not scale up from individual-scale processes and patterns, but the fact that you look at climate variables with temporal resolutions of at least 1,000 years means you are not addressing individual-scale processes (though I'm not 100% certain that this is the temporal resolution used; this information is not clearly presented in the Methods section). Further supporting this is that you are analyzing multiple species and lineages together over a million years.

*We agree with the reviewer that the labelling was not fully appropriate for these two scales, and also that our entire paper is generally interested in “evolutionary scale” changes. We changed the terms to “lifetime” and “lineage” to emphasize the shorter-scale (natural selection, adaptation) and longer-scale (fitting with the concept of “extinction filters” on lineage levels) which is at the conceptual heart of these hypotheses. As such the distinction remains relevant from an **analytical** perspective as different variables need to be examined to test the different hypotheses and we feel that previous work at times has been confusing as these scales are mixed. We further clarified this part by rewriting the section taking into account the reviewers’ comments (lines 100-112).*

5) Brain and body size variables should be log-transformed as these increase multiplicatively, not additively. Also, why are your climate variability variables log-transformed?

We thank the reviewer for pointing out this issue. As response to this comment, we now use the log-transformed versions of body and brain sizes in the main text and re-did all of our analyses accordingly. Where necessary, we refer to the natural-unit version of the analysis, which is presented in the supplement.

The log-transformation for climate variability variables was done purely for practical purposes. In the paper, we define climate variability (e.g., for precipitation) as the running variance over a 10ka window. To do so, we calculate the standard deviation for 10 time steps (1ka each). This can vary over several orders of magnitude, depending on the variable under consideration (e.g., see Figure 3d for the running 10ka standard deviation of mean annual precipitation).

6) Is it possible for the authors to upload their climate data? The analyses are not replicable nor completely transparent without these data. Likewise, I assume some coding language was used for the power analyses. If so, the code should be uploaded as well.

We thank the reviewer for pointing out this important omission of the previous manuscript version. The source code and the data to run all analyses in this paper can be accessed via https://osf.io/smyac/?view_only=fdb4be1f95f84acea103e8b302d26418 (view-only for the review process; we will add a DOI with the final code & data upon acceptance).

Specific comments:

1. Keywords: Shouldn’t “brain size” be included here?

Added “brain size” to the keywords.

2. Line 56: Are these “qualitative and verbal assessments”? It seems to me that latitude, global oxygen isotope curves, etc. are quantitative variables.

In the papers we are referring to, either latitude has been used as direct environmental proxy (which is not correct) or brain/body sizes have been associated with particularly OIS stages or plotted coarsely on global oxygen isotope curves. We do not find that these procedures qualify as quantitative assessment of the relation between environmental variables and body/brain sizes. We rephrased this section to make our assessment clearer.

3. Line 61-62: I am not sure that this was the main takeaway of Maxwell et al.’s paper. They argued that the hominin record is not sampled well enough to correlate those data with climate patterns. Furthermore, they did not investigate or even comment on hominin body and brain size.

We agree that this was not the main takeaway message of Maxwell et al’s paper. We rephrased this section to make our assessment clearer. We also take up some of the points in the Maxwell et al paper in the Methods section.

4. Line 73: “old world” should be capitalized.

Changed to “Old World”.

5. Line 74: Are *H. naledi* and *H. floresiensis* included in your analyses? They don't appear to be according to Figure 1. If they're not included, why were they excluded?

Brain and body size estimates from Homo floresiensis and Homo naledi were excluded from this analysis as we cannot use these taxonomic units to study the effects of environmental variables on body/brain size. H. floresiensis as a taxonomic unit has only produced one body and brain size estimate from a single site and age and so does not feature any environmental variation to study (no paleoanthropologist would assign it to any of the other taxonomic units we used here, and there is not even close to a consensus of assigning it to any of the existing evolving lineages). Principally the same goes for Homo naledi which is so-far known from one time slice and one site, although more size estimates are available. If more sites with either taxon would have been found this would definitely merit inclusion in this paper on studying the effects of environmental variables on body/brain size.

6. Table 1: Why temperature of the coldest quarter and precipitation of the driest quarter instead of warmest and wettest, respectively? This reasoning for this decision should be made explicit for the reader. And why isn't precipitation of the driest quarter analyzed for the Environmental Constraints Hypothesis?

For this paper we wanted to re-formulate and streamline the main hypotheses put forward for environmental variables driving brain and body size in Homo (from the literature as cited on page 4 of our manuscript). While the reviewer is correct in pointing out that more combinations of factors (and thus more hypotheses) are possible, here we are interested in a set of clearly formulated hypotheses that can be tested with appropriate environmental variables that are relevant (i.e. have been proposed, see page 4 literature) for human evolution - e.g. cold and dry has frequently been proposed as driver of larger body/brain size but not the other way around (e.g. hotter rather associated with smaller body size). Since we are interested in increases in the hominin lineage, we used these variables. As in comment #3 above, the text was adjusted to be explicit about this hitherto missing reasoning.

We fully agree with the reviewer on the second issue and added precipitation of the driest quarter to the analyses of the Environmental Constraints Hypothesis and adjusted the power and fossil analyses accordingly.

7. Line 151: "additive" and "linear" are redundant.

In response to the reviewer's comments, we have completely revised the naming of the linear models we have used throughout the analysis. The additive linear model (as referred to in Eq (2) in the Methods) accounts for the climate effect on (same slope for all taxa), whereas the second model, which we previously called interactive linear model, accounts for the climatic effects on the different taxonomies (different slopes for different taxa). While we use both models for the analysis of the fossil records, we only apply the additive linear model for the power analysis.

We have used better names for the different linear models throughout the text now:

The null model is referred to as LM-T (linear model for taxonomy only); the two alternative ones are LM-C (linear model with single climate effect), and LM-CT (linear model with taxa-specific climate effect).

8. Lines 169-170: A citation is needed here.

MK & AM: 0.8 is the commonly accepted threshold for acceptable power (e.g. used in NIH and other international grants). It is hard to come up with a citation that is not a standard textbook, as it is analogous to 0.05 as the significance (alpha) level.

9. Line 200: How did you determine "detected an association"? According to the Table 2 caption, "A model is highlighted in bold, if it has the strongest support (no highlight for a given variable means that the null taxonomic model is the most supported)". Did you use AIC to determine this? Was this determined from your power analyses (though the reader cannot connect your empirical results to the

power analyses results because you do not present slopes from the latter; see my point under “General Comments”)?

We thank the reviewer for this comment and see that we have not been clear enough about how we detect associations of body/brain size with climate variables. As noted by the reviewer, we use the difference in the AIC to detect the association. A difference smaller than -2 indicates that the alternative model has stronger support than the null (taxonomy-only) model. And no, we determine the AIC directly from the models for the fossil data; the power analysis results are not used here. The reviewer correctly points out that we miss to connect the results to our power analysis. We added clarification to the section “Power analysis and linear regressions” the captions of Table 2 and 3 and the results. We also added the hypothesized slopes for the “weak”, “medium”, and “strong” relationships as a table in the supplement (SOM Table 3) and refer to it in the analyses of the fossil data.

10. Lines 202-203: I would be careful with your wording here. A “null” result does not mean same or equal to zero. It just means you do not currently have a large enough sample size and statistical power to detect a difference or non-zero effect.

We could not find the respective passage in our text but updated the general methods and results and hope that this answers the current point.

11. Lines 200-210: According to Figure 2a, these results appear to be driven entirely by Late Pleistocene *H. sapiens*, so I would be careful about generalizing these findings to the other two taxonomic groups (this especially pertains to your Discussion). The slopes may be large for these other taxa (Table 2), but I bet the R2 are very low. (In fact, all this could be said of 2b, and the results don't appear generalizable to Neanderthals in 2c and 2d).

We disagree with the reviewer's interpretation for Figure 3a (mistakenly given as 2a by the reviewer) and the effect of MAT on the other two taxa. As the reviewer correctly concludes the slopes for both Neanderthals and MP Homo indicate a similar strong negative association with MAT (see Table 2, first row). To clarify the reviewer's concern about the R2, we have applied the linear model for each taxon separately. The resulting R2 (median and 95% CI) was 0.16 [0.05-0.27] for MP-Homo, 0.32 [0.16-0.44] for Neanderthals, and 0.17 [0.09-0.27] for Pleistocene Homo sapiens (compared to the additive model for all taxa: $R^2 = 0.22$ [0.16-0.29]). If anything, it is Neanderthals that show a clearer association with MAT (the same applies for Figure 3b). For the reviewer's comment on the association with NPP and MAPvar10 (Figures 3c and 3d), the results cannot be generalized to any taxa because the (LM-CT) model combines the differentiated effects of the three taxa. Only when combined into one model (i.e., via the interactive terms of taxa and NPP/MAPvar10) do we find that the model is better than the null model (LM-T).

12. Line 225: I find this result surprising, given that the interaction model increased R2 by 0.04 on average. The percentage in [] is certainly much higher, but again, what this means is difficult to interpret given the ambiguity surrounding what “most supported” means.

The percentages in [] indicate how many times out of the 100 thinned runs the (LC or LCT) model performs better than the null model in terms of AIC differences. The increase of the R2 indicates how much more variance in the body/brain size can be explained if a relationship with the climate variable is included. In general, a model that includes a climate variable is the better model if it has a lower AIC despite the penalty (of -2 AIC points) for adding another factor to the regression model. We added this clarification also to the main text and table captions to avoid confusion. Compared to body size/MAT, this result is more moderate in terms of its effect on brain size.

13. Lines 232-233: What was the median and 95% range calculated on? The 100 thinned datasets? This should be made clear.

Yes, the median and 95% ranges are based on the 100 thinned datasets. We have added this in the respective table captions.

14. Tables 2 & 3: The addition of an interaction term only results in non-negligible increases in model R2. You also did not do power analyses for interaction terms (or at least did not present the results), so it is not possible to interpret your interaction results in that light. Given all this, I would say the interaction terms did not add anything to your models.

In order to add the interaction terms to the power analysis, we would have to hypothesize those in terms of their effect size (as we did for the “weak”, “medium”, and “strong”). For the more general additive model, we have had a good starting point based on previous studies of Bergmann’s rule. Based on this number, we derived an objective classification into “strong” (1/4 of total possible slope), “medium” (1/8), and “weak” (1/16). This would have not been possible for the interaction terms. The added value of the interaction terms is to analyse whether there is an association with climate within each taxon. Neanderthals and Mid-Pleistocene Homo for example, have mostly lived in very different environments. Therefore, one would expect different evolutionary responses to those different environments and/or at different evolutionary stages (i.e. different slopes). For example, Pleistocene Homo sapiens shows no association with NPP whereas Mid-Pleistocene Homo and Neanderthals do. The additive model does not differentiate between the different taxa and thus loses these details if they did indeed exist.

15. Tables 2 & 3: I would scale at least your climate variables to be SD=1 (you can scale body and brain size so the slopes are comparable between these two dependent variables). It is difficult to compare the linear model slopes across climate variables, given that the variables are measured in such different units.

We have thought about standardizing the datasets for the reason the reviewer has pointed out. The main argument against it is that we include the coefficient of variation (CV) as an uncertainty estimate for both body and brain size. The CV depends on the mean and can therefore not be applied to a standardized data set (which has a mean of zero). One advantage of using actual units is that we can compare our results (e.g., Bergmann’s rule) to existing literature (which usually list measured values in real units). Furthermore, any future study that might use a different data set (e.g., with more fossil records) can make direct comparisons and there is no need to (un-)standardize any results from this study.

16. Lines 263-264: I would not interpret this as a real result given the negligible increase in R2 relative to the null model (Table 3).

We disagree with the assessment by the reviewer here. The relatively small increase in R2 is due to the fact that taxonomic brain size differences are already relatively large (which is different to the null model, LM-T in body size which is weaker; see also Figure 1). As such, our alternative models (LM-C, LM-CT) run against a very hard competition of a null model for brain size particularly. We added this now also to the text to clarify this point, which might also be what the reviewer found lacking here (lines 218-226). However, the additional effect of NPP within MP Homo and Neanderthals can be robustly inferred from the fossil record. In fact, if we remove Late Pleistocene Homo sapiens from the analysis, the null model R2 is 0.35 (0.31-0.40) whereas the interactive R2 is 0.45 (0.39-0.52). This is an increase of 10 percent points in terms of variance explained. We also added some of this clarification to the results section (lines 218-226).

17. Lines 291-295: Then why wasn’t an effect found? It could be that the effect is not there (as you propose) or instead that the observed effect is weak, or that your chosen error values in the power analyses are not matching what is found in your dataset. The slope itself can tell you whether the effect is non-existent, weak, or strong (the slope is unbiased and will give you the right answer on average even in the face of noisy data, though the standard errors will be higher).

We agree with the reviewer that a comparison between the hypothesized and measured effect sizes would indeed be helpful to discuss differences between the power and the fossil data analysis. We thus added SOM Table 3 and added some of this information to the results section.

18. Line 353: Should this be cm³?

We thank the reviewer for pointing out this mistake and changed it to cm³.

19. Line 359: Why were they useful? It should be stated here and not left to the reader to find the reason in another publication.

We rewrote this section to make the decision for the three taxonomic groupings clearer.

20. Line 426: Shouldn't the intercept be equal to Y minus the slope times X? You have it the other way around.

We thank the reviewer for pointing out this mistake, which needs to be given the other way round. We have fixed this issue and now added a full section on our linear regression and updated the exact equations for the three different models.

21. Lines 427-430: Age uncertainties are always published along with estimated ages. Why not use those instead of assuming some arbitrary value?

For this revision we now provide the age uncertainties along with the estimated ages in SOM1 and SOM2 wherever available. This being said, age uncertainties are not available for all of the fossil specimens. In addition, there are sometimes conflicting ages (e.g. older C14 vs. newer AMS C14 ages; different estimation methods) and we had to find a reasonable assessment of these ages (as now explained in SOM1 and SOM2). As this makes it impossible to treat the entire dataset in the same way, we choose to use set values that approximate the true age uncertainties. In most cases, our age uncertainties fit well with the published age uncertainties though for many our calculations are even larger (!) than the published ones, providing more conservative age estimates and thus higher robusticity of our results.

22. Line 430: Why is temperature given a fixed error of 2K when all other variables get a percentage error? This results in a smaller error for temperature than your other climate variables. I wonder if this is why the only linear models that have non-negligible R² are those related to temperature in the body size analyses (Table 2)

23. Lines 430-431: Why these values of error instead of some other number? These numbers are not justified nor are there cited references. Therefore, these numbers seem kind of arbitrary.

Comment for #22 and #23: We added a sentence in the Methods section as to why we choose the 2°C for temperature variables (MAT, BIO11) and 20% for all other variables. In terms of the units, we followed the convention to express the model error for temperature in units of °C (for example, see Fifth Assessment Report — IPCC; Chapter 9, Fig. 9.2c; <https://www.ipcc.ch/report/ar5/wg1/evaluation-of-climate-models/fig9-02-2/> and <https://www.ipcc.ch/report/ar5/wg1/evaluation-of-climate-models/fig9-04-2/>) and model errors for precipitation in % (for example, Fifth Assessment Report — IPCC; Chapter 9, Fig. 9.4d; <https://www.ipcc.ch/report/ar5/wg1/evaluation-of-climate-models/fig9-02-2/> and <https://www.ipcc.ch/report/ar5/wg1/evaluation-of-climate-models/fig9-04-2/>). The 2°C is a compromise of the mean absolute errors (MAE) of HadCM3 for the periods of the Holocene (~6ka BP, MAE=1.1°C), the Last Glacial Maximum (~21ka BP, MAE=3.0°C) and the Last Interglacial (~125ka BP, MAE=2.0°C), and derives from a recent study that evaluates the model biases in HadCM3, i.e., the same model that the climate reconstructions for this study are based on (Beyer et al., 2020). From the same paper, we choose 20% for MAP (and BIO17) as a compromise for the MAE as calculated for the Holocene (MAE=12%) and the Last Glacial Maximum (38%). For simplicity, we choose the same

20% as an appropriate model error estimate because no detailed references for those variables (NPP and the log of the 10ka running standard deviations of all climate variables).

24. Lines 431-435: What is a “population of a taxonomic unit of Homo”? Did you somehow spatiotemporally circumscribe populations within each species? Furthermore, was the CV calculated across all 19 hunter-gatherer populations? Is this estimate applicable to hominin taxa (potentially multiple species within mid-Pleistocene Homo) measured over evolutionary time? Surely, there is more variation in the latter. The same criticism applies to the brain size dataset used. Quickly looking at the brain size dataset from Du et al. (2018), six H. heidelbergensis specimens have brain sizes with a CV of 35%.

We changed wording in this section (i.e. our used values are used for taxonomic units and should be considered as “estimates” of the variability as it is empirically unknown) and now provide a clearer explanation and references for the used values for body size and brain size. We also stress that in paleontological samples, due to low sample sizes and the effects of time-averaging (particularly in samples that have accumulated over many tens of thousands of years) variance is artificially inflated compared to neontological ones.

25. Lines 435-437: This seems kind of “brute force”. I would just include taxonomy as a separate variable and let the model estimate the different intercepts for each taxonomic group, i.e., $Y = \beta_0 + \beta_1 * Taxon + \beta_1 X$.

This correction is only for the generation of the synthetic data sets. We want each data set to be as close to the fossil data sets in terms of its mean and its variance. As response to this comment we have expanded the paragraph accordingly and provide the respective equations. They can now be found in lines 425-444 with detailed descriptions for each.

26. Lines 462-464: Did you create a version of Figure 2 for the interaction models? If you didn't, why not? Based on the linear model results (Tables 2 and 3), I assume none of those percentages comparing the interaction to the additive model would be high (cf. Figure 2).

We did not do so as we had not done a power analysis for the interactive model. As explained in our reply to comment #14, the power analysis would have had too many additional terms, i.e., effects, which we cannot possibly assess in a realistic way. The interactive model would require us to hypothesize two more interaction terms (for two out of the three taxa at least). The sheer number of synthetic data sets to be created, based on those additional interaction terms, would be too large (it scales with $O(n^k)$, with n being the number of synthetic data sets, i.e., 1000 and k being the number of hypothetical effects, e.g., 3).

27. References: genus and species names should be italicized.

We italicized accordingly.

Reviewers' Comments:

Reviewer #1:

Remarks to the Author:

I consider that the authors have addressed all the significant issues raised by reviewers that they practically can. As they acknowledge, some of the issues raised are beyond the power of the available data to properly address or resolve, while other issues would take the paper into areas beyond those that they are trying to address in this paper.

I consider that the paper is now suitable for publication, though I did have a few more small suggested edits/issues in the attached document.

I do not require anonymity.

Chris Stringer

Reviewer #3:

Remarks to the Author:

I thank the authors for thoughtfully considering the reviewers' comments and incorporating them accordingly. What follows are some remaining concerns I have.

General comments:

1. I appreciate the thoughtful consideration of scale in your paper, but it seems to me that the distinction between "lifetime" and "lineage" (cf. Table 1) does not apply to your analyses. According to your Figure 3, you are looking at body/brain size as a function of some climate variable WITHIN lineages. This means that all your analyses are looking at the same scale, namely natural selection within a species over many generations. It is not clear to me how extinction (your proposed lineage-level phenomenon) is investigated in your analysis. Focusing on the Environmental Variability Hypothesis, your second sentence (Lines 136-138) implies selection for larger brains/bodies within lineages and does not need to invoke extinction (e.g., a more variable environment would select for a larger-brained, smarter organism to deal with resource unpredictability). Therefore, I would get rid of the distinction between "lifetime" and "lineage" scales. Your Table 1 is fine without it, as these hypotheses look at different environmental variables anyway.

Moreover, I think the use of "lifetime" is incorrect. Natural selection operates over multiple generations, not within the lifetime of an individual (which is a very Lamarckian way of looking at things). I think "generational" would be a more appropriate term, but this suggestion is rendered moot by my previous one, recommending that you remove the lifetime/lineage scale distinction.

2. I understand that a power analysis is used to assess whether a significant relationship between the dependent and independent variable(s) is detectable (i.e., $p < 0.05$), given a certain sample size, effect size, and variation around that effect size. One can hold two of these constant and vary the third to see under what conditions a significant relationship can be detected. My original criticism was centered on the fact that a power analysis seems unnecessary here because a significance test between the observed dependent and independent variables is ITSELF a power analysis: if $p < 0.05$, then you have enough statistical power to detect the effect, and the opposite is true for $p > 0.05$ (or $\Delta AIC < 2$, which is pretty much the same as $p > 0.05$; see

<https://dynamicecology.wordpress.com/2015/05/21/why-aic-appeals-to-ecologists-lowest-instincts/> and the comments section). This is why a power analysis is traditionally done BEFORE data collection to see how large of a sample size is needed to detect an effect, assuming one exists (though in biology, there is ALWAYS an effect, but it can be miniscule). As a result, I am not sure what is learned from doing a power analysis varying effect size, especially when the true effect size is unknown and cannot be altered. A power analysis may tell you that you can only detect an effect if the effect size is large, but a test of the relationship between your two observed variables will essentially tell you that

also ($p < 0.05$ means the effect is large enough, and $p > 0.05$ means it is not; and because the real effect size cannot change, one can only await more fossil discoveries to increase sample size and statistical power if $p > 0.05$). All of this is supported by your Lines 448-450, which describes a P-value, not a power analysis. I think your power analysis adds extra layers to your analysis that encumbers its clarity and comprehension for readers.

That said, I do think there is tremendous value in what you are doing in terms of exploring how dating, climate, and brain/body size measurement error affect your results. To me, this is more of a sensitivity analysis, which I wholeheartedly support. The most straightforward way to do this would be to resample age, climate, and brain/body size within their error ranges, fit your linear model, and assess the coefficient estimates and their significance. Repeat this 1,000 times to see how these sources of error introduce variation in your statistical inferences. I would strongly recommend this approach over your power analysis as currently structured.

Specific comments:

Line 1: I think "explain" is probably more accurate than "predict". Prediction involves more than just a regression on a single observed dataset (i.e., cross-validation).

Line 78: I would provide more details about what an "individual data point" is. Is it a site for a given time period?

Lines 141-143: I think a citation is needed here. Is it actually true that more variable environments select for smaller body sizes? I am not aware of any research showing that this is the case.

Line 144: I would be careful with the wording here. The environmental variables are not proxies for the mechanisms of interest (e.g., larger body/brain size being selected for in colder environments). They are just measuring the environmental variables of interest in your hypotheses.

Line 165: Why is null italicized? This also applies to italicized words in Lines 426-427. I do not think the italics are necessary.

Line 198: But your power analyses show that you do not have enough statistical power to detect this relationship, so why is this considered to be a "real" relationship? I think this example demonstrates how it is not clear how your power analyses shed light on the interpretation of your empirical results.

Line 200: Lower by how much? By two?

Lines 214-216: How was it determined that the association was valid for two hominin taxa but not the third?

Lines 216-218: As before, your power analysis shows that your statistical analyses should not be able to detect this effect. How was it determined that this association is "real", and does that mean your power analyses are not informing how you interpret your empirical results? If so, what is the purpose of the power analyses?

Lines 224-226: This is true, but this is also the purpose of a null model. That is, you want to see how much variance is explained by climate on top of differences attributed to taxonomic differences. Therefore, to me, the explanation here does not discount the fact that climate does explain relatively little variance in the dependent variable.

Tables 2 & 3: I would recommend using the actual climate variable name instead of the Bioclim abbreviations, so readers do not have to flip back and forth between these tables and Table 1.

Line 239: Difference of how much? (Applies to Table 3 as well)

Lines 242-243: I find this to be an odd way to present R2. I would present the actual model R2 but also include the R2 for the null model in the caption as you have done. (Applies to Table 3 as well)

Line 251: Should this be "brain size"?

Lines 258-260: It is not clear to me what the "30-yr averages" are. For each 1,000-year time period, you averaged 30 years' worth of data? Why 30? How are the 30 years distributed within each 1,000-year period?

Line 274: Your study does not address questions of plasticity, given the large time scale of your analyses. To get at plasticity, you would need to see how body size changes within the lifetime of a single fossil hominin individual.

Lines 309-310: This is stated too strongly. It could be that the true effect sizes are weak, which your power analyses show cannot be detected. On a more philosophical note, it is likely that there is no such thing as a true negative in biology. Effect sizes are never exactly zero, and no two groups are exactly the same. Therefore, $p > 0.05$ just means one's sample size is not large enough, and p will always be < 0.05 with a large enough sample size. Dushoff et al. 2019 in *Methods Ecology and Evolution* make this point nicely.

Lines 403-407: I think the grammar of this sentence needs to be edited for clarity.

Line 427: Recommend changing to "The null model simply estimates the mean for each taxonomic group..."

Line 431: "modifier" is unclear here, and I would clarify that the estimated coefficient for each taxon is the estimated deviation of the mean of the taxon of interest from the baseline taxon.

Lines 513-515: Lower by how much?

Table 2: MAT and BIO11 are bolded and thus inferred to explain body size, but BIO17 and NPP are not bolded. What does it mean then if only two of the climate variables in the Environmental Stress Hypothesis explain body size? Is this hypothesis supported, partially supported, or not supported? It is too late now, but it would have been useful to a priori determine what constitutes a strong test of this hypothesis, concerning which and how many climate variables need to explain body size. This same criticism applies to the results from Table 3.

Table 3: NPP is bolded even though only 76% of the 100 thinned datasets support this climate variable. What is the percentage cut-off used to determine whether a climate variable is bolded or not? This is not clear from the caption.

Reply to reviewers

We found the critical and constructive comments of the two reviewers to be extremely helpful during revision of our manuscript and appreciate the time and effort that they put into the review process. We agree with the majority of the reviewers' comments and have endeavoured to address each of them carefully in the revised contribution, to the extent that we can. This included mostly modifications to the main text, a better explanation of how the power analysis was used in this study and addition of the suggested sensitivity analysis as part of the main analysis of fossil data. Importantly, the main results and conclusions of this study remain unchanged.

In the reply to reviewers below we provide a more detailed description of how the criticisms and suggestions have been addressed point by point. Reviewers' comments are in normal font, our responses in *italic*.

Reviewer #1 (Remarks to the Author):

1) I consider that the authors have addressed all the significant issues raised by reviewers that they practically can. As they acknowledge, some of the issues raised are beyond the power of the available data to properly address or resolve, while other issues would take the paper into areas beyond those that they are trying to address in this paper.

I consider that the paper is now suitable for publication, though I did have a few more small suggested edits/issues in the attached document.

We thank the reviewer again for his positive overall assessment as well as the helpful corrections as suggested in the attached document. We incorporated all of these changes (e.g., spelling, italicizing species and genus names in Figure 3).

Reviewer #3 (Remarks to the Author):

General comments:

1. I appreciate the thoughtful consideration of scale in your paper, but it seems to me that the distinction between "lifetime" and "lineage" (cf. Table 1) does not apply to your analyses. According to your Figure 3, you are looking at body/brain size as a function of some climate variable WITHIN lineages. This means that all your analyses are looking at the same scale, namely natural selection within a species over many generations. It is not clear to me how extinction (your proposed lineage-level phenomenon) is investigated in your analysis. Focusing on the Environmental Variability Hypothesis, your second sentence (Lines 136-138) implies selection for larger brains/bodies within lineages and does not need to invoke extinction (e.g., a more variable environment would select for a larger-brained, smarter organism to deal with resource unpredictability). Therefore, I would get rid of the distinction between "lifetime" and "lineage" scales. Your Table 1 is fine without it, as these hypotheses look at different environmental variables anyway. Moreover, I think the use of "lifetime" is incorrect. Natural selection operates over multiple generations, not within the lifetime of an individual (which is a very Lamarckian way of looking at things). I think "generational" would be a more appropriate term, but this suggestion is rendered moot by my previous one, recommending that you remove the lifetime/lineage scale distinction.

While we still maintain that it is important to consider that we are dealing with more than one temporal scale in our analyses, we take the ongoing concerns of the reviewer on this issue seriously. Our original intention was to create less and not more confusion on the different time scales involved. Seeing that this is not the case - and considering potential problems with our chosen terminology – we have removed the strong distinction between “lifetime” and “lineage” scales from Table 1 and the text. Instead, we refer to short- and long-term time scales in the text.

2. I understand that a power analysis is used to assess whether a significant relationship between the dependent and independent variable(s) is detectable (i.e., $p < 0.05$), given a certain sample size, effect size, and variation around that effect size. One can hold two of these constant and vary the third to see under what conditions a significant relationship can be detected. My original criticism was centered on the fact that a power analysis seems unnecessary here because a significance test between the observed dependent and independent variables is ITSELF a power analysis: if $p < 0.05$, then you have enough statistical power to detect the effect, and the opposite is true for $p > 0.05$ (or $\Delta AIC < 2$, which is pretty much the same as $p > 0.05$; see <https://dynamicecology.wordpress.com/2015/05/21/why-aic-appeals-to-ecologists-lowest-instincts/> and the comments section). This is why a power analysis is traditionally done BEFORE data collection to see how large of a sample size is needed to detect an effect, assuming one exists (though in biology, there is ALWAYS an effect, but it can be miniscule). As a result, I am not sure what is learned from doing a power analysis varying effect size, especially when the true effect size is unknown and cannot be altered. A power analysis may tell you that you can only detect an effect if the effect size is large, but a test of the relationship between your two observed variables will essentially tell you that also ($p < 0.05$ means the effect is large enough, and $p > 0.05$ means it is not; and because the real effect size cannot change, one can only await more fossil discoveries to increase sample size and statistical power if $p > 0.05$). All of this is supported by your Lines 448-450, which describes a P-value, not a power analysis. I think your power analysis adds extra layers to your analysis that encumbers its clarity and comprehension for readers.

That said, I do think there is tremendous value in what you are doing in terms of exploring how dating, climate, and brain/body size measurement error affect your results. To me, this is more of a sensitivity analysis, which I wholeheartedly support. The most straightforward way to do this would be to resample age, climate, and brain/body size within their error ranges, fit your linear model, and assess the coefficient estimates and their significance. Repeat this 1,000 times to see how these sources of error introduce variation in your statistical inferences. I would strongly recommend this approach over your power analysis as currently structured.

The main role of the power analysis is to allow the reader to assess how informative negative results are. The key issue is whether a non-significant result can be interpreted as informative (i.e., we had sufficient power to detect a pattern but we did not) or uninformative (we did not have enough power, and thus a non-significant result tells us little). We have further clarified the main Methods text (lines 165-172) describing the power analysis to explain how we used it in this specific case (see also the Methods at the end of the document).

The referee also makes the really helpful suggestion of performing a sensitivity analysis. We have now extended our approach by adding the uncertainty of dating and climatic variables (we had already done that for the power analysis, so it does make a lot of sense to include these two sources of uncertainty into the main analysis to reach consistency across the two). Rather than producing a sensitivity analysis to be put in the supplementary, we have replaced our main analysis of the fossil data with a version that includes these additional uncertainties,

as it provides a more robust assessment of the patterns. While individual figures (e.g., percentage values and R²-values) are slightly different, the main results remain unchanged.

Specific comments:

Line 1: I think “explain” is probably more accurate than “predict”. Prediction involves more than just a regression on a single observed dataset (i.e., cross-validation).

We feel the term “explain” is rather loaded, as it can be interpreted as implying a mechanism. For this reason, we feel that the term “predict” is more appropriate in this context, as it emphasises that we can use a variable to predict what happens to a measurement, but we do not imply direct causation.

Line 78: I would provide more details about what an “individual data point” is. Is it a site for a given time period?

*We clarified the sentence accordingly: “The environmental information for each individual data point (i.e., **geographical location and age of each fossil specimen**) comes from a climate emulator (27) that takes into account long-term, glacial-interglacial climate variation, caused by changes in the Earth’s orbit around the sun (Milankovitch cycles) (28) and in greenhouse gases, such as CO₂.”*

Lines 141-143: I think a citation is needed here. Is it actually true that more variable environments select for smaller body sizes? I am not aware of any research showing that this is the case.

The four hypotheses guiding our paper are based on a synthesis from the previous literature on body and brain sizes in primates and hominins. All relevant citations can be found in the literature discussion before (lines 100-103) and all of these processes have been proposed for brain and/or body size. Note also that none of the specific hypotheses here have individual citations for the reasons above.

Line 144: I would be careful with the wording here. The environmental variables are not proxies for the mechanisms of interest (e.g., larger body/brain size being selected for in colder environments). They are just measuring the environmental variables of interest in your hypotheses.

In statistics, the definition of a proxy variable is one that is itself not the direct causal variable, but it replaces an unobservable or immeasurable variable. As such, we do feel that we are correct in our use of the term: temperature of the coldest quarter is a proxy variable for the selection pressure exerted by cold climate on body size.

Line 165: Why is null italicized? This also applies to italicized words in Lines 426-427. I do not think the italics are necessary.

We removed the italics at these places.

Line 198: But your power analyses show that you do not have enough statistical power to detect this relationship, so why is this considered to be a “real” relationship? I think this example demonstrates how it is not clear how your power analyses shed light on the interpretation of your empirical results.

A power analysis tells us about our confidence that we can find a pattern if there is one. We can find a pattern even if we have low initial power, but we have to be lucky. The problem with low power is that, if we do not find a pattern, we do not know whether one is there as we might have simply missed it. A power of 0.8 tells us that we have an 80% chance of finding a pattern of a given magnitude if it existed. So, we can find a pattern even with very low power.

Line 200: Lower by how much? By two?

Yes, by an AIC difference of 2. The alternative model should have at least an AIC that is 2 points higher than the null model (the AIC penalizes models with additional parameters). We have added this information to the main Methods text and the Methods at the end of the manuscript. See also the more detailed comment and reply below.

Lines 214-216: How was it determined that the association was valid for two hominin taxa but not the third?

We looked at the confidence interval of the taxon-specific slopes; if they included zero, there was no detectable effect for that taxon. The relevant slopes and CIs are found in the Table(s) 2 and 3.

Lines 216-218: As before, your power analysis shows that your statistical analyses should not be able to detect this effect. How was it determined that this association is “real”, and does that mean your power analyses are not informing how you interpret your empirical results? If so, what is the purpose of the power analyses?

See the response above, this seems to be a misinterpretation of what a power analysis does. A power of 0.2 would mean that we have a chance of 20% of finding a signal if it was there, so we can still have significant results with low power.

Lines 224-226: This is true, but this is also the purpose of a null model. That is, you want to see how much variance is explained by climate on top of differences attributed to taxonomic differences. Therefore, to me, the explanation here does not discount the fact that climate does explain relatively little variance in the dependent variable.

We agree that climate explains a low amount of variation for brain size but maintain that it is important to exactly assess how much. This also does not negate our general findings of different environmental drivers of body and brain size.

As a response to the comment by the reviewer, however, we added to the discussion the following statement in lines 311-319: “We did find relationships with the 10ka-sigma of mean annual precipitation and net primary productivity (NPP), but the variance in brain size explained by these variables was small compared to the effect of MAT on body size. These results suggest that brain size is less influenced by environmental variables in Homo during the last 1.0 Ma compared to body size. Apart from other drivers being likely more relevant (see below), one factor contributing to the difficulty of detecting environmental effects lies in the strong performance of the null model (LM-T) based on taxonomic differences in brain size variations which explained much more variance ($R^2=0.47$) compared to body size ($R^2=0.05$).” and lines 377-379: “We also note that many of the environmental variables provided no detectable correlations and explained variance is often low, raising doubts about an unquestioned a priori reliance on environmental factors in explaining macro-processes in

human evolution”. We also adjusted our abstract accordingly, emphasizing that the variance explained by climate for body size is higher than for brain size.

Tables 2 & 3: I would recommend using the actual climate variable name instead of the Bioclim abbreviations, so readers do not have to flip back and forth between these tables and Table 1.

We followed the reviewer’s suggestions and used the actual climate variable name instead of the Bioclim abbreviations in Tables 1-3.

Line 239: Difference of how much? (Applies to Table 3 as well)

We have added that the AIC difference needs differ by at least 2 in Tables 2 and 3.

We also added the following clarification to the methods section (lines 182-190): “We then compared the explanatory power of these models using the Akaike Information Criterion (AIC) (37), estimating the difference in AIC between the alternative models and the null model. A positive difference ($\Delta AIC > 0$) implies that the alternative model is the better model, but we chose a more conservative AIC difference of 2 to yield more robust results. The power of our analysis in recovering a relationship of a given strength between size and a climatic variable was then defined as the proportion of datasets for which a (hypothetical) climatic effect could be detected (i.e., the AIC values of LM-C and LM-CT are equal or larger than 2 compared to the AIC of the null model, LM-T). “

We emphasize here that this difference of 2 points is generally considered a quite conservative assessment when assessing the different performance of models (with 2 points meaning a significant outperformance) and we are thus confident in the robusticity of our findings.

Lines 242-243: I find this to be an odd way to present R2. I would present the actual model R2 but also include the R2 for the null model in the caption as you have done. (Applies to Table 3 as well)

We think this the best way to present results from an ensemble of models. Each of our 1000 models (based on the thinning and randomizing the independent variables) has its own R2. To summarize them thoroughly we presented the median R2 (from all models) as well as the 95-percentile range.

Line 251: Should this be “brain size”?

We thank the reviewer for spotting this error and corrected it.

Lines 258-260: It is not clear to me what the “30-yr averages” are. For each 1,000-year time period, you averaged 30 years’ worth of data? Why 30? How are the 30 years distributed within each 1,000-year period?

We haven’t calculated them ourselves, but this is a common procedure to provide paleo-climate simulation data (otherwise the amount of data to process and download would be too large). The original HadCM3 simulation results are provided as climatological means and these are calculated from the last 30 years of the respective HadCM3 simulations (Singarayer

and Valdes, 2010). The idea is that these climatologies are then representative of each 1000-yr snapshot period.

Singarayer, Joy S., and Paul J. Valdes. 2010. 'High-Latitude Climate Sensitivity to Ice-Sheet Forcing over the Last 120kyr'. *Quaternary Science Reviews* 29 (1): 43–55.
<https://doi.org/10.1016/j.quascirev.2009.10.011>.

Line 274: Your study does not address questions of plasticity, given the large time scale of your analyses. To get at plasticity, you would need to see how body size changes within the lifetime of a single fossil hominin individual.

We would disagree with the reviewer here. From our study, we cannot fully confirm the exact causality behind the changes in body sizes we see in our study (though we know it is stemming from thermal stress). To do so would be to reach beyond our data. As such, we think it is more adequate to leave this question open and state the two processes which could be the driver - also potentially in combination - behind this pattern, namely either natural selection and/or phenotypic plasticity.

Lines 309-310: This is stated too strongly. It could be that the true effect sizes are weak, which your power analyses show cannot be detected. On a more philosophical note, it is likely that there is no such thing as a true negative in biology. Effect sizes are never exactly zero, and no two groups are exactly the same. Therefore, $p > 0.05$ just means one's sample size is not large enough, and p will always be < 0.05 with a large enough sample size. Dushoff et al. 2019 in *Methods Ecology and Evolution* make this point nicely.

We changed our text accordingly (lines 352-355): "The synthetic data thus suggest that our negative results for these variables, and the lack of support for the Environmental Stress and the Environmental Constraints Hypothesis, are either "true negative" findings or that true effect sizes are relatively small."

Lines 403-407: I think the grammar of this sentence needs to be edited for clarity.

We changed the text accordingly: "The main idea behind GCMET is that global climate model (GCM) simulations of the last 120,000 years contain sufficient information about long-term climatic changes on time scales of 1,000 years and longer. Given that we know the external boundary conditions, we can reconstruct previous glacial-interglacial climatic changes."

Line 427: Recommend changing to "The null model simply estimates the mean for each taxonomic group..."

We have changed the sentence as suggested.

Line 431: "modifier" is unclear here, and I would clarify that the estimated coefficient for each taxon is the estimated deviation of the mean of the taxon of interest from the baseline taxon.

We have clarified that β_1 is a factor associated with the deviation from the reference taxon's mean size

Lines 513-515: Lower by how much?

Lower by 2 points. We have added the AIC difference of 2 to the text (see reply to comments above).

Table 2: MAT and BIO11 are bolded and thus inferred to explain body size, but BIO17 and NPP are not bolded. What does it mean then if only two of the climate variables in the Environmental Stress Hypothesis explain body size? Is this hypothesis supported, partially supported, or not supported? It is too late now, but it would have been useful to a priori determine what constitutes a strong test of this hypothesis, concerning which and how many climate variables need to explain body size. This same criticism applies to the results from Table 3.

Table 3: NPP is bolded even though only 76% of the 100 thinned datasets support this climate variable. What is the percentage cut-off used to determine whether a climate variable is bolded or not? This is not clear from the caption.

We have now streamlined the results part in terms of which variables are bolded and changed the table captions accordingly. For the fossil data analyses, we have bolded a variable if the LM-C or the LM-CT provides a better model (by an AIC difference of 2) among the 1000 runs compared to the LM-T, indicating a climate effect. We have bolded either the LM-C or LM-CT if it is the best model among the three. Our approach in the results highlights and discusses the individual percentages, providing a transparent assessment of how often the climate models are found to have an effect. In addition, we also report and discuss the R^2 -values of these models. From both the percentages and the R^2 -values it is clear that for body size there is a strong support for MAT and BIO11, whereas there is some support for BIO17 and NPP, but weaker compared to body size. We also evaluate these results in more detail now in the discussion (see reply above). We also qualify and discuss the implications of our results for specific hypotheses in more detail now (i.e., in how much they support or don't support them). For example, we now clearly state in our discussion that only some of the predictions of the Environmental Stress Hypothesis apply, i.e. with temperature (i.e. thermal stress) being the key driver (and not low precipitation) and added (lines 309-310) "We failed to detect any effect of low rainfall or nutrient-poor environments as determinants of stress in our analyses."

Reviewers' Comments:

Reviewer #3:

Remarks to the Author:

While I still disagree with the authors regarding the rationale for a power analysis, I appreciate their willingness to engage with me on this issue. It seems we have reached an impasse, so I will not comment on the power analysis any further. What remains are minor comments on my end.

General comments:

I found the model acronyms to be slightly non-intuitive. LM-T for Equation 1 makes sense to me since taxon is the only predictor in the model. However, LM-C does not indicate a linear model wherein climate is included as the only predictor (taxon is included as well; Eq. 2). I would therefore be inclined to call this model LM-CT. For the model with the interaction term (Eq. 3), perhaps that can be called LM-CxT or LM-C*T.

Specific comments:

Lines 36-38: Why would a low proportion of variance explained by environmental factors mean environment still had an (indirect) influence? Why not conclude that environment had little to do with brain size evolution?

Lines 206-209: I like the framing here of how one should interpret negative results. I would add that it could also be that the variation surrounding the effect size is larger than originally assumed. This also applies to Lines 352-355.

Line 222: How can a fixed value of 0.57kg be associated with a percent increase of 0.87%? Won't the kg value associated with a 0.87% increase depend on what kg value you start off with? This applies to Line 226 also.

Line 232: Missing "for" after "accounted".

REVIEWERS' COMMENTS (comments/modifications by authors in *italics*)

Reviewer #3 (Remarks to the Author):

While I still disagree with the authors regarding the rationale for a power analysis, I appreciate their willingness to engage with me on this issue. It seems we have reached an impasse, so I will not comment on the power analysis any further. What remains are minor comments on my end.

General comments:

I found the model acronyms to be slightly non-intuitive. LM-T for Equation 1 makes sense to me since taxon is the only predictor in the model. However, LM-C does not indicate a linear model wherein climate is included as the only predictor (taxon is included as well; Eq. 2). I would therefore be inclined to call this model LM-CT. For the model with the interaction term (Eq. 3), perhaps that can be called LM-CxT or LM-C*T.

*We agree with the reviewer and suggest the following improvements to the naming of the alternative models. Starting with the null model (LM-T) the alternative models form a hierarchy, where more terms are added. Therefore, we think the naming can and should reflect that. We have now changed the model with same climate effect for all taxa to **LM-TC** (was LM-C). The second model with the inclusion of an interaction of each taxon with climate is now referred to as **LM-T*C** (was LM-CT). This new naming also reflects how statistical programs such as R implement linear models (the * denotes an interaction between two variables).*

Specific comments:

Lines 36-38: Why would a low proportion of variance explained by environmental factors mean environment still had an (indirect) influence? Why not conclude that environment had little to do with brain size evolution?

There seems to be a misunderstanding here. We do not state that the indirect effect is inferred from the low proportion of variance (we also note that the proportion of explained variance is low but not negligible, and thus does not warrant a conclusion that the environment had little to do with brain size evolution). The speculation about indirect effects comes from the nature of the variables associated with brain size, as we spell out in the discussion (line 272-283).

Lines 206-209: I like the framing here of how one should interpret negative results. I would add that it could also be that the variation surrounding the effect size is larger than originally assumed. This also applies to Lines 352-355.

It is certainly correct that, if the real amount of variability was larger than what we used for the power analysis, we might have had less power than we estimated. However, we did consider relatively generous estimates of variability for the different components, and accounted for their combined effects, so we feel that our estimates were actually likely to be on the conservative side.

Line 222: How can a fixed value of 0.57kg be associated with a percent increase of 0.87%? Won't the kg value associated with a 0.87% increase depend on what kg value you start off with? This applies to Line 226 also.

We had attempted to present effect sizes for both the analysis on log values (proportions) and raw values (a fixed amount in kg), but we realise that this is confusing. Thus, we removed the 0.57kg change per degree of cooling reference (this result is based on the same analysis but done for the untransformed body size data, i.e., in units of kg) and replaced it with a simple example what that

effect size means in practical terms: "For example, a 2°C cooling in MAT would be associated with a body size increase of 1 kg for individuals with a weight of 60 kg ($0.87\%/^{\circ}\text{C} \times 2^{\circ}\text{C} \times 60\text{kg} = 1\text{kg}$)."

We removed this for coldest quarter temperature.

Line 232: Missing "for" after "accounted".

We corrected this spelling error.